# High light and temperature reduce photosynthetic efficiency through different mechanisms in the C$_4$ model *Setaria viridis*

Cheyenne M. Anderson[1,10], Erin M. Mattoon [1,2,10], Ningning Zhang[1], Eric Becker[1], William McHargue [1], Jiani Yang [1], Dhruv Patel [3], Oliver Dautermann [3], Scott A. M. McAdam [4], Tonantzin Tarin [5,6], Sunita Pathak [1], Tom J. Avenson[7], Jeffrey Berry[1], Maxwell Braud [1], Krishna K. Niyogi [3,8,9], Margaret Wilson[1], Dmitri A. Nusinow [1], Rodrigo Vargas [5], Kirk J. Czymmek [1], Andrea L. Eveland [1] & Ru Zhang [1✉]

C$_4$ plants frequently experience high light and high temperature conditions in the field, which reduce growth and yield. However, the mechanisms underlying these stress responses in C$_4$ plants have been under-explored, especially the coordination between mesophyll (M) and bundle sheath (BS) cells. We investigated how the C$_4$ model plant *Setaria viridis* responded to a four-hour high light or high temperature treatment at photosynthetic, transcriptomic, and ultrastructural levels. Although we observed a comparable reduction of photosynthetic efficiency in high light or high temperature treated leaves, detailed analysis of multi-level responses revealed important differences in key pathways and M/BS specificity responding to high light and high temperature. We provide a systematic analysis of high light and high temperature responses in *S. viridis*, reveal different acclimation strategies to these two stresses in C$_4$ plants, discover unique light/temperature responses in C$_4$ plants in comparison to C$_3$ plants, and identify potential targets to improve abiotic stress tolerance in C$_4$ crops.

[1] Donald Danforth Plant Science Center, St. Louis, MO, USA. [2] Plant and Microbial Biosciences Program, Division of Biology and Biomedical Sciences, Washington University in Saint Louis, St. Louis, MO, USA. [3] Department of Plant and Microbial Biology, University of California, Berkeley, CA, USA. [4] Purdue Center for Plant Biology, Department of Botany and Plant Pathology, Purdue University, West Lafayette, IN, USA. [5] Department of Plant and Soil Sciences, University of Delaware, Newark, DE, USA. [6] Instituto de Ecología, Universidad Nacional Autónoma de México, Mexico City, Mexico. [7] Department of Plant Sciences, University of Cambridge, Cambridge, UK. [8] Howard Hughes Medical Institute, Berkeley, CA, USA. [9] Molecular Biophysics and Integrated Bioimaging Division, Lawrence Berkeley National Laboratory, Berkeley, CA, USA. [10] These authors contributed equally: Cheyenne M. Anderson, Erin M. Mattoon. ✉email: rzhang@danforthcenter.org

Several of the world's most economically important staple crops utilize C₄ photosynthesis, including *Zea mays* and *Sorghum bicolor*. C₄ photosynthesis concentrates CO₂ around Rubisco (ribulose-1,5-bisphosphate carboxylase/oxygenase) by employing biochemical reactions within mesophyll (M) and bundle sheath (BS) cells[1,2]. The high local concentration of CO₂ near Rubisco favors carbon fixation over photorespiration, which is initiated by the oxygenase activity of Rubisco[1,3]. C₄ photosynthesis is hypothesized to have been selected by low CO₂, high light (HL), and high temperature (HT) conditions[4,5]. C₄ plants typically exhibit higher photosynthetic and water-use efficiencies than their C₃ counterparts under high light or high temperature[6]. However, C₄ crops experience more frequent, damaging high light or high temperature stresses in their natural environments than C₃ crops, with reduced C₄ crop yield regularly occurring in warmer regions[7]. As the mean global temperature continues to increase, maize yields are estimated to decrease between 4% and 12% for an increase of each degree Celsius[7]. Photosynthesis in maize leaves is inhibited at leaf temperatures above 38 °C. Recent data from 408 sorghum cultivars shows that breeding efforts over the last few decades have developed high-yielding sorghum cultivars with considerable variability in heat resilience and even the most heat-tolerant sorghum cultivars did not offer much resilience to warming temperatures, with a temperature threshold of 33 °C, beyond which sorghum yields start to decline[8]. Under natural conditions, especially at the tops of canopies, direct sunlight can be very intense and thus oversaturate the photosynthetic mechanism in C₄ plants. Sorghum leaves had reduced stomatal conductance and net CO₂ assimilation rates after 4 h exposure to high light mimicking nature sunlight[9]. To improve C₄ crop yields, it is crucial to holistically approach how C₄ plants respond to high light or high temperature, two of the most influential environmental factors that can compromise C₄ photosynthesis.

High light responses have been studied extensively in C₃ plants[10–15]. To cope with reactive oxygen species (ROS) production and photooxidative stress resulting from high light, C₃ plants have evolved many protective mechanisms which act on different timescales[10,14]. Non-photochemical quenching (NPQ), especially its predominant component, energy-dependent quenching (qE), acts within seconds to dissipate excess light energy as heat[10,16]. The formation of qE depends on the thylakoid lumen pH, the photosystem II (PSII) polypeptide PsbS, and the accumulation of the xanthophyll pigment zeaxanthin[17–19]. In C₃ plants, under high light, violaxanthin is converted to the intermediate pigment antheraxanthin which is then converted to zeaxanthin by the enzyme violaxanthin de-epoxidase[20]. Accumulation of zeaxanthin is also necessary for the induction of a slower-relaxing component of NPQ, zeaxanthin-dependent quenching (qZ)[21]. State transitions, which restructure the light-harvesting complexes (LHCs) around PSII and PSI, occur on the order of minutes[10,16]. When photoprotective processes are insufficient, high light can result in photoinhibition (qI), which takes hours to recover[10]. Following high light exposure, expansion of the thylakoid lumen, swelling of the grana margin, and de-stacking of the thylakoid grana facilitate PSII repair by promoting accessibility and repair of PSII machinery[15,22–24]. High light stress also results in dynamic transcriptional regulation of photosynthetic genes and induces the abscisic acid (ABA) pathway in the C₃ model plant *Arabidopsis thaliana* (Arabidopsis throughout)[11].

High temperature is known to affect many cellular processes in C₃ plants, including various aspects of photosynthesis[25–29]. C₃ plants under high temperature have shown decreases in photosynthetic rates, inactivation of Rubisco, reduction of plastoquinone (PQ), and increase in cyclic electron flow (CEF) around photosystem I (PSI)[30]. Arabidopsis leaves treated with

high temperature of 40 °C had increased plastoglobuli (PG) formation[31]. PG are thylakoid-associated plastid lipoprotein particles whose size, shape, and counts respond to abiotic stresses[32]. Additionally, high temperature induces the expression of heat shock transcription factors (HSFs), many of which have been implicated in transcriptional responses to numerous abiotic stresses, including high light and high temperature[33]. The induced HSFs increase the expression of heat shock proteins (HSPs), which are chaperone proteins involved in proper protein folding in response to high temperature and other abiotic stresses[34].

Unlike C₃ plants, studies on how C₄ plants respond to high light or high temperature are largely limited, especially the underlying coordination between mesophyll and bundle sheath cells and the multi-level effects of high light and high temperature on photosynthesis, transcriptomes, and ultrastructure of C₄ plants. A recent study examined the effects of high light stress in the C₄ grass *Setaria viridis* over 4 days, with sampling points for photosynthetic parameters, sugar quantification, and transcriptome analyses every 24 h[35]. They reported relatively minor transcriptional changes but a large accumulation of sugars without repression of photosynthesis in high-light-treated samples[35]. These results suggest that leaves with prolonged high-light treatment undergo adaptive acclimation and transcriptional homeostasis in a few days. However, the short-term transcriptional responses of C₄ plants to high light remain largely unknown. In sorghum leaves, high light induced the avoidance response in mesophyll chloroplasts and the swelling of bundle sheath chloroplasts (by cross-section light microscope images), but the underlying mechanisms are unclear[9]. Research about how C₄ photosynthesis responds to high temperature is mainly limited to biochemical and gas exchange analyses which suggest that high temperature results in Rubisco activation[36], affects the activities of C₄ carbon fixation enzymes[37], and decreases the bundle sheath conductance while increases CO₂ leakiness[38,39]. Two transcriptome analyses in maize under high temperature have been reported[40,41], but thorough analysis of C₄ transcriptome with multi-level effects under high temperature is rare. Additionally, ultrastructural analysis in C₄ plants under high light or high temperature can help us understand how these two stresses limits C₄ photosynthesis and affects the coordination between mesophyll and bundle sheath cells, but currently such information is lacking.

To gain deeper insights into the molecular and physiological responses of C₄ plants to high light or high temperature, we used the green foxtail *Setaria viridis* as a model. *S. viridis* is an excellent model to study C₄ photosynthesis because of its expanding genetics and genomics toolkit, relatively quick generation time (8~10 weeks, seed to seed), and similarity to agronomically important C₄ crops, e.g., maize and sorghum[2,42,43]. We hypothesized that high light or high temperature affected C₄ plants at different levels and linking multi-level changes could improve our understanding of high light or high temperature tolerance in C₄ plants. We investigated the response of *S. viridis* to moderately high light or high temperature over a 4 h time course at photosynthetic, ultrastructural, and transcriptomic levels (Fig. 1a). We monitored the dynamic changes of transcriptomes, pigments, and ABA levels during the different treatments. We also measured photosynthetic parameters and ultrastructural changes after 4 h treatments, which revealed cumulative changes associated with the different treatments.

Although we observed a comparable reduction in photosynthetic efficiency in high-light- or high-temperature-treated leaves, detailed analysis at multiple levels revealed different acclimation strategies to these two stresses in *S. viridis*. The transcriptional changes under high temperature were less extensive and more

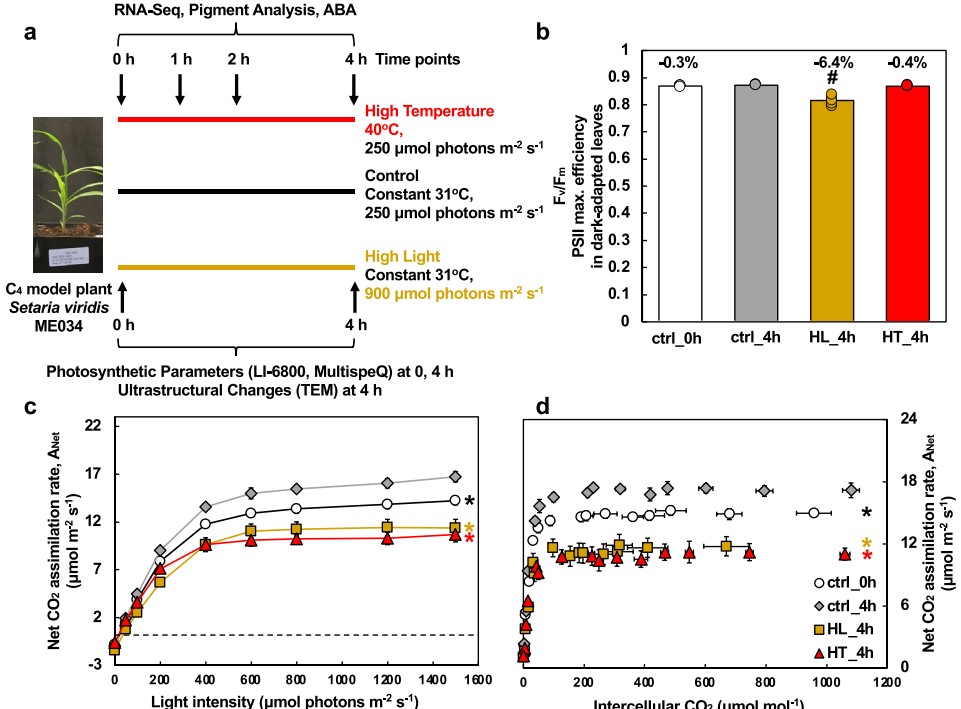

**Fig. 1 High light and high temperature resulted in a comparable reduction in net $CO_2$ assimilation rates and high light also caused significant photoinhibition in *S. viridis* leaves. a** Experimental overview. We investigated how the $C_4$ model plant *S. viridis* ME034 responded to high light or high temperature at different levels. Plants grown under the control condition were treated with control condition or high light or high temperature for 4 h. Leaf tissues from different treatments were harvested at different time points for the analysis of RNA-seq, pigments, and leaf ABA levels. Photosynthetic parameters were measured at 0 and 4 h time points, including gas exchange and chlorophyll fluorescence using LI-6800 and spectroscopic measurements using MultispeQ. Transmission electron microscopy (TEM) analysis was performed to investigate chloroplast ultrastructure changes in leaves after 4 h treatments. **b** High-light-treated leaves had reduced PSII maximum efficiency ($F_v/F_m$) measured by chlorophyll fluorescence with 20 min dark-adapted leaves. Pound symbols indicate statistically significant differences of ctrl_0h (at the start of treatments), HL_4h (after 4 h HL), and HT_4h (after 4 h HT) compared to ctrl_4h (after 4 h control treatment) using Student's two-tailed t-test with unequal variance ($^\#p < 0.01$). Percentages indicate reduction in $F_v/F_m$ compared to ctrl_4h. **c, d** Net $CO_2$ assimilation rates during light response and $CO_2$ response, respectively. Most data points of ctrl_0h, HL_4h, and HT_4h were statistically significantly different compared to ctrl_4h using Student's two-tailed t-test with unequal variance, denoted by asterisks at the end of curves. p-Values were corrected for multiple comparisons using FDR ($^*p < 0.05$, the colors of * match the significance of the indicated conditions, black for ctrl_0h, yellow for HL_4h, red for HT_4h). Mean ± SE, $n = 3$–6 biological replicates.

dynamic than under high light. We revealed different responses of mesophyll and bundle sheath cells under high temperature or high light. The high-light-treated leaves had over-accumulated starch in both mesophyll and bundle sheath chloroplasts, which may increase chloroplast crowdedness and inhibit PSII repair. While both high light and high temperature induced PG formation in chloroplasts, high-temperature-treated mesophyll chloroplasts also had swollen grana. Additionally, we demonstrated the crosstalk between high light response and ABA signaling in $C_4$ plants. Our research provides a systematic analysis of high light and high temperature responses in *S. viridis* and identifies potential targets to improve stress tolerance in $C_4$ crops.

## Results

### High light or high temperature caused a comparable reduction in photosynthesis and high light also resulted in photoinhibition. *S. viridis* leaves treated with 4 h high light (HL_4h) exhibited significantly reduced maximum efficiencies of PSII ($F_v/F_m$) as compared to those with 4 h control treatment (ctrl_4h) (Fig. 1b), suggesting high-light-induced photoinhibition. Net $CO_2$ assimilation rates ($A_{Net}$) were significantly reduced in high-light- or high-temperature-treated leaves in response to changes in light or $CO_2$ (Fig. 1c, d). Pre-treatment control leaves (ctrl_0h) also had lower $A_{Net}$ as compared to ctrl_4h leaves, suggesting circadian regulation of photosynthesis over the course of the day. The

comparisons between different treatments at the 4 h time point should exclude the effects of circadian regulation. Leaf temperature was stable at 31 °C under control and high light treatments while it increased gradually from 31 to 37 °C by the end of 4 h treatment of 40 °C (Supplementary Fig. 1). Stomatal conductance and transpiration rates in response to light were reduced in HL_4h leaves, especially at the beginning of the light response curve (Supplementary Fig. 2a, c). Stomatal conductance and transpiration rate in response to $CO_2$ were lower in HL_4h or HT_4h leaves than in ctrl_4h leaves (Supplementary Fig. 2b, d). PSII efficiency and electron transport rates in light-adapted leaves were reduced in HL_4h leaves as compared to ctrl_4h leaves in response to light (Supplementary Fig. 2e, g).

To estimate and model a variety of photosynthetic parameters, we assessed various aspects of leaf-level gas exchange measurements based on the light response curves and $CO_2$ response curves (Supplementary Fig. 3). High light or high temperature compromised photosynthetic capacities and reduced several photosynthetic parameters in HL_4h and HT_4h leaves compared to ctrl_4h leaves, including gross maximum $CO_2$ assimilation rates ($A_{max}$), maximum carboxylation rates ($V_{cmax}$), and quantum yields of $CO_2$ assimilation (Supplementary Fig. 3a–c). HL_4h leaves had reduced stomatal conductance ($g_s$) but increased light compensation point as compared to ctrl_4h leaves (Supplementary Fig. 3e, g). HT_4h leaves had reduced light

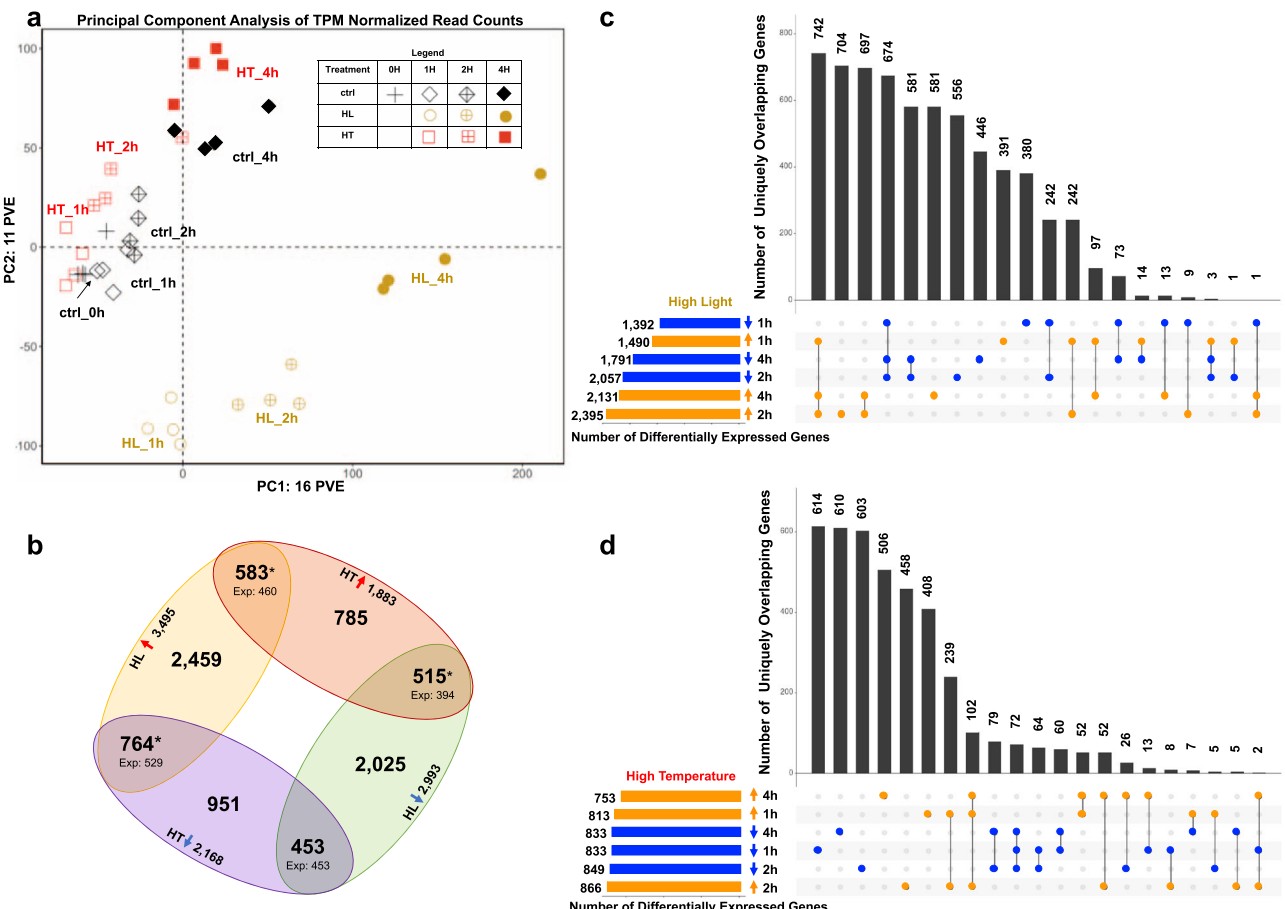

**Fig. 2 Time-course transcriptome data reveal dynamic responses to high light or high temperature in *S. viridis*. a** Principal component analysis of TPM (transcripts per million) normalized read counts in control, high-light-, and high-temperature-treated samples. The first two principal components (PC) representing the highest percent variance explained are displayed. PC1 explains 16% of the variance in the dataset and mainly separates the samples based on time. PC2 explains 11% of the variance in the dataset and mainly separates the high light samples from the control and high temperature samples. Black diamonds indicate control samples, yellow circles indicate high light samples, and red squares indicate high temperature samples. Different fillings for these symbols indicate different time points of each treatment. Each treatment and time point have four biological replicates, represented by symbols with the same shape and color. **b** High light and high temperature treatments had more overlapping differentially expressed genes than expected by random chance. Gene sets represent the number of genes differentially regulated in at least one time point in the given condition. Red upward arrows denote up-regulation and blue downward arrows denote down-regulation. Yellow oval denotes high light up-regulated genes, green oval denotes high light down-regulated genes, red oval denotes high temperature up-regulated genes, purple oval denotes high temperature down-regulated genes. Expected values (Exp) are the number of the overlapping genes expected by random chance based on size of the gene lists and background of all genes tested via DeSeq2 (14,302). Numbers above expected values are the actual number of overlapped genes between two conditions. *$p < 0.0001$, Fisher's exact test. **c, d** High temperature transcriptional responses are more transient than high light. UpSetR plots show number of uniquely overlapping genes between up- and down-regulated gene sets at each time point in high light and high temperature, respectively. Horizontal bars indicate the number of genes up- or down-regulated at each time point. Filled circles indicate the gene sets included in the overlap shown. Vertical bars indicate the number of genes represented in the overlap shown. Overlapping gene sets are arranged in descending order by number of genes. Genes may only belong to a single overlapping gene set and are sorted into the overlapping set with the highest number of interactions.

saturation point as compared to ctrl_4h leaves (Supplementary Fig. 3h).

**Transcriptomics revealed important differences in key pathways responding to high light or high temperature**. To investigate the transcriptional patterns that may underlie the photosynthetic phenomena observed above, we performed RNA-seq analysis (Fig. 1a). Principal component analysis (PCA) of transcripts per million (TPM) (Supplementary Data 1) normalized read counts from control, high light, and high temperature treatments showed that the experimental conditions dominated the variance in the dataset (Fig. 2a).

Next, we compared differentially expressed genes (DEGs) between high light and high temperature treatments. Genes that

were either up- or down-regulated in at least one time point were included in the lists of DEGs for each condition. Utilizing this method, we were able to broadly compare the trends between the high light and high temperature transcriptomes. There were more DEGs identified in the high light dataset than in the high-temperature dataset (Fig. 2b, Supplementary Table 1, and Supplementary Data 2). Significantly more genes were up- or down-regulated in both high-light- and high-temperature-treated transcriptomes than would be expected by random chance (Fig. 2b and Supplementary Data 4). Additionally, significantly more genes were regulated in opposite directions between high light and high temperature transcriptomes than would be expected by random chance. To visualize how DEGs were conserved between time points within treatments, we plotted the

overlaps between up- and down-regulated genes at each time point. In high-light-treated samples, 742 genes were up-regulated at 1, 2, and 4 h time points, representing the largest subset of uniquely overlapping genes and the core high-light-induced genes (Fig. 2c and Supplementary Data 3). Similarly, 674 genes were down-regulated at all the three time points of high light treatment, representing the core high-light-reduced genes. Conversely, in the high-temperature-treated samples, the expression pattern was dominated by genes differentially expressed at a single time point (Fig. 2d), indicating that the transcriptional response to high temperature was more transient and dynamic than that to high light. In high-temperature-treated samples, 102 and 72 genes were up- and down-regulated at all the three time points, representing the core high-temperature-induced and -reduced genes, respectively.

To reveal transcriptional changes that may explain the reduced photosynthesis under high light or high temperature, we grouped DEGs into several key pathways. Investigation of genes related to the light reaction of photosynthesis revealed that many genes involved in PSII assembly/repair and photoprotection (e.g., *PsbS*) were up-regulated in high light, while many genes relating to LHCII and the core complexes of PSII/PSI were down-regulated in high light (Fig. 3a, b). Although high temperature treatment did not result in the same extent of differential regulation of light-reaction-related genes as high light, STN7, a kinase involved in state 1 to state 2 transitions[44] was induced, while TAP38, a phosphatase involved in state 2 to state 1 transitions[45] was repressed in high-temperature-treated leaves (Supplementary Fig. 4a). This suggests a possible heat-induced state transition to move the mobile LHCII from PSII (state 1) to PSI (state 2). Additionally, several genes related to the chloroplast NDH (NADPH dehydrogenase) complex were up-regulated in the high temperature treatment (Fig. 3b). Furthermore, when investigating genes involved in CEF (Supplementary Fig. 4), we found that key components of CEF, *PGR5 (proton gradient regulation 5)*[46] and two copies of *PGRL1 (PGR5-like photosynthetic phenotype 1)*[47], were induced under high temperature, suggesting heat-induced CEF around PSI.

Under high light treatment, the transcriptional changes of genes involved in the Calvin-Benson cycle were less extensive than those involved in the light reactions of photosynthesis (Fig. 3c). Rubisco activase (RCA) is essential for $CO_2$ fixation by maintaining the active status of Rubisco[48,49]. The *S. viridis* genome has two adjacent genes encoding RCAs (Supplementary Fig. 5). Protein sequence alignment of the two *S. viridis* RCAs with Arabidopsis RCAs revealed one SvRCA-α that retains the two conserved redox-sensitive cysteine residues as in *AtRCA_α*, and one SvRCA-β that has a higher basal expression (approximately 700-fold higher) than *SvRCA_α* and possibly the major RCA in *S. viridis*. *SvRCA_α* was highly induced during the entire 4 h high temperature treatment (Fig. 3c).

Key genes involved in photorespiration, e.g., GOX1 (glycolate oxidase)[50,51] and AGT1 (Serine:glyoxylate aminotransferase)[52] were down-regulated under high light (Fig. 3c). GOX1 and several other genes involved in photorespiration, PGLP1 (2-phosphoglycolate phosphatase)[53], HPR1 (hydroxypyruvate reductase)[54], and PLGG1 (plastidic glycolate/glycerate transporter)[55] were induced under high temperature, suggesting heat-induced photorespiration.

Some genes important for $C_4$ carbon metabolism were up-regulated under high light (Fig. 3c), e.g., PEPC_B (phosphoenylpyruvate carboxylase) and NADP-MDH (NAD-dependent malate dehydrogenase)[1]. Carbonic anhydrase[56] (CA_A) was induced under both high light and high temperature.

By investigating pathways associated with photosynthesis, we found high light increased the expression of starch biosynthesis/degradation genes and genes encoding PG-localized proteins

(Fig. 4a), but down-regulated several genes in the sugar-sensing pathway (Fig. 4b) and differentially regulated several sugar transporter genes (Supplementary Fig. 4b). These transcriptional changes were much less pronounced under high temperature.

Several HSFs had highly induced expression under either high light or high temperature, but interestingly, different HSFs were up-regulated during these two stresses (Fig. 4c). HSFA6B was a notable exception, which was induced in both high light and high temperature. A set of shared HSPs were induced under both stresses, but the induction was quicker and stronger under high temperature than under high light, especially the small HSPs, suggesting shared and also temporally distinct transcriptional responses of HSPs under high light and high temperature.

We also investigated genes associated with ROS pathways (Supplementary Fig. 4c). Specialized ROS scavenging pathways have evolved in plants[57]. We identified genes encoding antioxidant enzymes in *S. viridis* and investigated their expression patterns under high light or high temperature. Three gene families of antioxidant enzymes have many members with strong differential expression in high-light-treated leaves: *TRX* (thioredoxin), *POX* (peroxidases), and *GST* (glutathione S-transferase). Interestingly, within each of the three antioxidant pathways, some genes were up-regulated while others were down-regulated in high-light-treated leaves. A similar pattern was shown in high-temperature-treated leaves, although with fewer differentially regulated genes.

The reduced stomatal conductance in HL_4h leaves (Supplementary Fig. 2a) suggested there may be changes in ABA pathways and leaf ABA levels. Our RNA-seq analysis showed that several genes in the ABA pathways were up-regulated in response to high light (Fig. 5a). Additionally, ABA levels were increased 3-fold in HL_1h leaves followed by a return to baseline by HL_4h (Fig. 5b).

To distinguish mesophyll- and bundle sheath-specific transcriptomic responses and gain more information about how these two specialized cell types function together under high light or high temperature, we investigated the cell-type specificity of our pathways of interest (Supplementary Fig. 6 and Supplementary Data 6) using previously published mesophyll- and bundle sheath-specific transcriptomes under control conditions[58]. We observed several cell-type-specific transcriptional responses to high light or high temperature, e.g., pathways related to ROS scavenging, sugar transport, and HSPs.

**High light treatment induced NPQ in *S. viridis*.** The increased photoinhibition and *PsbS* transcription in HL_4h leaves prompted us to quantify NPQ and xanthophyll pigments. NPQ was significantly higher in HL_4h leaves than in ctrl_4h leaves in response to light and $CO_2$ (Fig. 6a, b). The high-light-induced NPQ measured by LI-6800 was confirmed using MultispeQ with the estimated NPQ, $NPQ_{(T)}$, based on a method that estimates NPQ in light-adapted leaves[59] (Supplementary Fig. 7c). The increased NPQ was also supported by the observed 4-fold increase of zeaxanthin (Fig. 6c) during high light. Additionally, high light treatment doubled the intermediate antheraxanthin level (Fig. 6d) and tripled the overall de-epoxidation state of the xanthophyll cycle (Fig. 6e). In Arabidopsis, lutein also has a role in NPQ or qE and can substitute for zeaxanthin in qE formation[60]. Lutein as well as total carotenoids were significantly induced in HL_4h leaves (Supplementary Fig. 8c, d). These results indicate the occurrence of photoprotection in high-light-treated leaves. Control and high temperature treatments had little effect on leaf pigments.

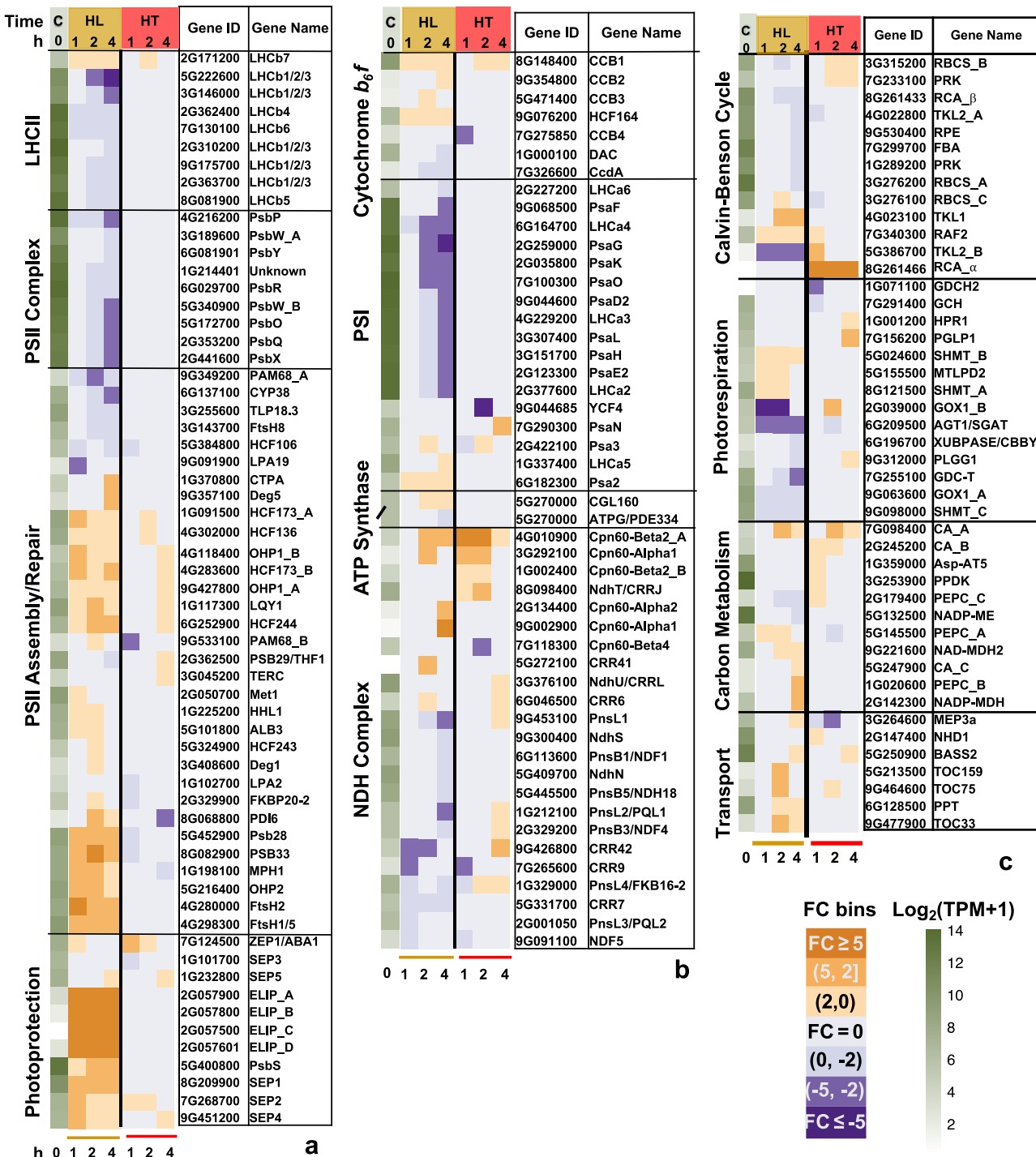

**Fig. 3 High light differentially regulated genes involved in photosynthesis more than high temperature. a**, **b** Genes related to light reaction of photosynthesis and photoprotection. **c** Genes related to carbon metabolism and chloroplast transport. The first green column displays log$_2$(mean TPM + 1) at ctrl_0h (at the start of treatments, C). TPM, transcripts per million, normalized read counts. Heatmap displays the fold change (FC) bin of DeSeq2 model output values at 1, 2, and 4 h of high light or high temperature versus control at the same time point (q < 0.05). FC bins: highly induced: FC ≥ 5; moderately induced: 5 > FC ≥ 2; slightly induced: 2 > FC > 0; not differentially expressed: FC = 0; slightly repressed: 0 > FC > −2; moderately repressed: −2 ≥ FC > −5; highly repressed: FC ≤ −5. Gene ID: *S. viridis* v2.1 gene ID, excluding "Sevir". All genes presented in the heatmaps were significantly differentially regulated in at least one time point.

**High light or high temperature altered chloroplast ultra-structures.** The reduced photosynthesis (Fig. 1c, d) in HL_4h and HT_4h leaves, and the high-light-induced photoinhibition (Fig. 1b) and transcripts related to the starch as well as PG pathways (Fig. 4a) led us to investigate the ultrastructural

changes of the mesophyll and bundle sheath chloroplasts in ctrl_4h, HL_4h, and HT_4h leaves using transmission electron microscopy (TEM) (Supplementary Fig. 9). TEM images showed HL_4h leaves had increased relative starch volume fraction and chloroplast area in both mesophyll and bundle sheath

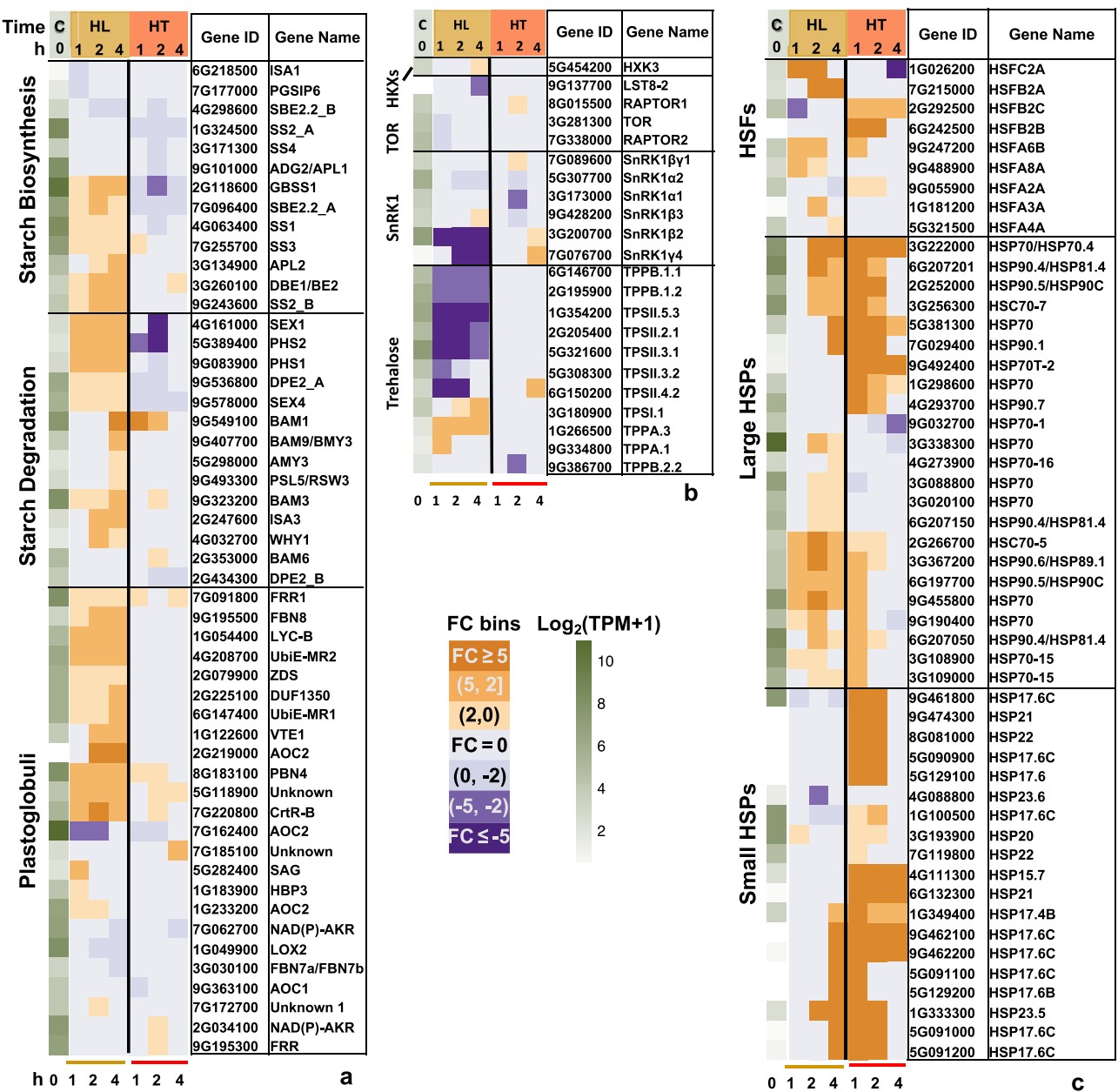

**Fig. 4 High light and high temperature differentially regulated genes involved in several key pathways. a**, **b** High light induced genes involved in starch biosynthesis/degradation and genes encoding plastoglobuli-localized proteins; **b** high light down-regulated many genes of the sugar-sensing pathways. **c** Both high-light- and high-temperature-induced genes encoding heat shock transcription factors (HSFs) and heat shock proteins (HSPs), but the induction was much quicker under high temperature than under high light. The first green column displays $\log_2$(mean TPM + 1) at ctrl_0h (at the start of treatments, C). TPM, transcripts per million, normalized read counts. Heatmap displays the fold change (FC) bin of DeSeq2 model output values at 1, 2, and 4 h of high light or high temperature versus control at the same time point ($q < 0.05$). FC bins: highly induced: FC ≥ 5; moderately induced: 5 > FC ≥ 2; slightly induced: 2 > FC > 0; not differentially expressed: FC = 0; slightly repressed: 0 > FC > −2; moderately repressed: −2 ≥ FC > −5; highly repressed: FC ≤ −5. Gene ID: *S. viridis* v2.1 gene ID, excluding "Sevir". All genes presented in the heatmaps were significantly differentially regulated in at least one time point.

chloroplasts, but decreased relative volume fractions of stroma plus stroma lamellae (unstacked thylakoid membranes) in mesophyll chloroplasts as compared to ctrl_4h leaves (Fig. 7 and Supplementary Table 2), suggesting increased starch accumulation and chloroplast crowdedness under high light. Starch quantification using biochemical assays confirmed 3x higher starch levels in HL_4h leaves as compared to ctrl_4h leaves (Fig. 7m). In HT_4h leaves, the relative starch volume fractions had a small increase in bundle sheath chloroplasts but decreased in mesophyll chloroplasts as compared to the control condition (Fig. 7 and Supplementary Table 2). Considering the small

reduction of bundle sheath chloroplast area, the starch volume either had little change or slightly reduced as compared to the control, consistent with the starch quantification (Fig. 7m). High temperature did not affect the relative volume of stroma or stroma lamellae in either mesophyll or bundle sheath chloroplasts (Fig. 7i).

Like in other $C_4$ plants, *S. viridis* grana are predominantly present in the mesophyll chloroplasts. Bundle sheath chloroplasts also have some grana, which are absent from the central area but present in the peripheral region (Fig. 7d–f). High light reduced grana width in mesophyll chloroplasts and the relative

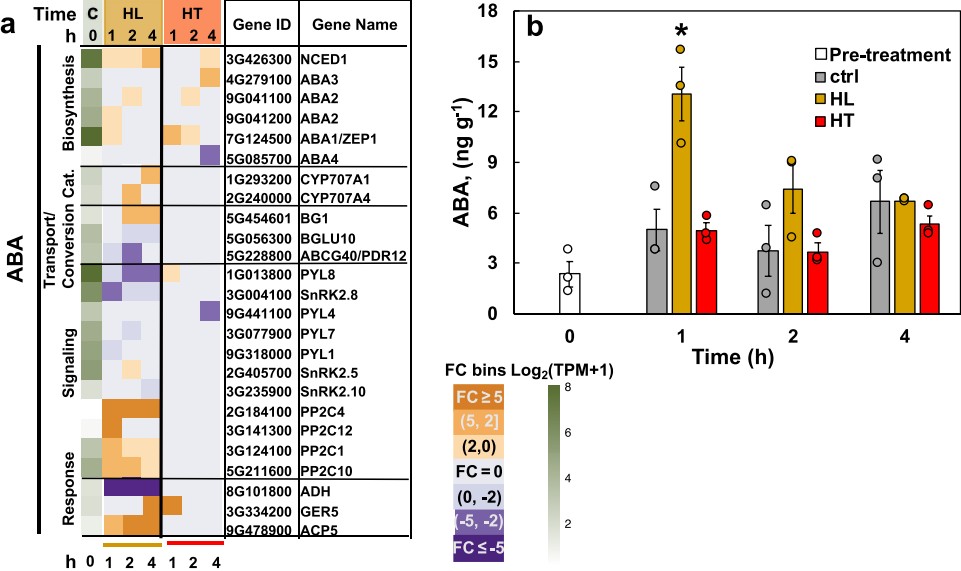

**Fig. 5 High light up-regulated genes involved in the abscisic acid (ABA) pathway and transiently increased leaf ABA levels. a** Heatmap of differentially regulated genes involved in the ABA pathway. Cat: catabolism. The first green column displays $\log_2$(mean TPM + 1) at ctrl_0h (at the start of treatments, C). TPM, transcripts per million, normalized read counts. Heatmap displays the fold change (FC) bin of DeSeq2 model output values at 1, 2, and 4 h of high light or high temperature versus control at the same time point ($q < 0.05$). FC bins: highly induced: FC ≥ 5; moderately induced: 5 > FC ≥ 2; slightly induced: 2 > FC > 0; not differentially expressed: FC = 0; slightly repressed: 0 > FC > −2; moderately repressed: −2 ≥ FC > −5; highly repressed: FC ≤ −5. Gene ID: *S. viridis* v2.1 gene ID, excluding "Sevir". All genes presented in the heatmaps were significantly differentially regulated in at least one time point. **b** Concentrations of leaf ABA. Mean ± SE, $n = 3$ biological replicates. Asterisk symbol indicates statistically significant differences as compared to the control condition at the same time point (Student's two-tailed *t*-test with unequal variance, *0.01 < $p$ < 0.05).

volume, height, and area of grana in bundle sheath chloroplasts as compared to the control condition (Fig. 7j, Supplementary Fig. 10, and Supplementary Table 2). The high temperature effects on grana structure were quite different from high light. Mesophyll chloroplasts under high temperature had increased relative volume, height, area, and mean layer thickness of grana, indicating heat-induced grana swelling. However, in bundle sheath chloroplasts, high temperature decreased the relative volume, width, and area of grana, suggesting that high temperature affected the grana structure differently in mesophyll and bundle sheath chloroplasts.

High light increased PG count and the total PG area per chloroplast, while it decreased the mean individual PG size in mesophyll chloroplasts, indicating smaller but more numerous PGs in mesophyll chloroplasts (Fig. 7k, l and Supplementary Table 2). Furthermore, high light increased individual PG size and total PG area per chloroplast in bundle sheath chloroplasts (Supplementary Table 2). High temperature increased individual PG size and total PG area, suggesting heat-induced PG formation in both mesophyll and bundle sheath chloroplasts.

**High-light- and high-temperature-treated leaves had reduced photosynthetic capacity.** The over-accumulated starch in HL_4h leaves (Fig. 7) and the increased leaf ABA levels (Fig. 5) led us to investigate photosynthesis immediately after different treatments without dark-adaptation under simulated stress conditions in the LI-6800 leaf chamber (Fig. 8). Under the same temperature and light intensity in the LI-6800 leaf chamber, most photosynthetic parameters with or without dark-adaptation were similar (groups 1 vs. 2) (Fig. 8). Under the simulated treatment condition in the LI-6800 leaf chamber (group 3), HL_4h leaves had higher net CO$_2$ assimilation rates ($A_{\text{Net}}$) and stomatal conductance ($g_s$) under 600 μmol photons m$^{-2}$ s$^{-1}$ light than ctrl_4h leaves under 200 μmol photons m$^{-2}$ s$^{-1}$ light, but both parameters in HL_4h

leaves were lower than those in ctrl_4h leaves under the same light intensity (groups 3 and 4) (Fig. 8a). This suggests that HL_4h leaves had reduced capacities for $A_{\text{Net}}$ and $g_s$ as compared to ctrl_4h leaves under the same condition. Under the simulated treatment condition (group 3), HL_4h leaves under 600 μmol photons m$^{-2}$ s$^{-1}$ light had reduced PSII operating efficiency (Fig. 8c), increased electron transport rates (Fig. 8d), and increased NPQ (Fig. 8f) as compared to the ctrl_4h leaves under 200 μmol photons m$^{-2}$ s$^{-1}$ light, consistent with light-induced electron transport and NPQ.

Without dark-adaptation, HT_4h leaves had similar $A_{\text{Net}}$ as ctrl_4h leaves (Fig. 8a, group 2). This may be due to the transient recovery of photosynthesis after switching the HT_4h leaves from 40 °C in the growth chamber to 25 °C in the LI-6800 leaf chamber for measurements. Under the same light intensity, HT_4h leaves had significantly lower $A_{\text{Net}}$ (Fig. 8a) and more reduced plastoquinone (Fig. 8e) than ctrl_4h leaves. Under the simulated treatment condition in LI-6800 leaf chamber (group 3), HT_4h leaves had increased stomatal conductance (Fig. 8b) but reduced $A_{\text{Net}}$ as compared to ctrl_4h leaves (Fig. 8a), consistent with transpiration cooling of leaf temperature (Supplementary Fig. 1) and reduced photosynthetic capacity in high-temperature-treated leaves (Supplementary Fig. 3a–c).

**The activity of ATP synthase was inhibited in high-light-treated leaves.** Based on the high-light-induced starch accumulation, we hypothesized that starch may inhibit photosynthesis through feedback regulation. We measured electrochromic shift (ECS) and chlorophyll fluorescence using MultispeQ[61] to evaluate proton fluxes and the transthylakoid proton motive force (*pmf*) in vivo[62–64]. Different treatments did not significantly change *pmf* (Fig. 9a). HL_4h leaves had significantly reduced proton conductivity and lower proton flux rates as compared to ctrl_4h leaves (Fig. 9b, c), indicating reduced ATP synthase activity in

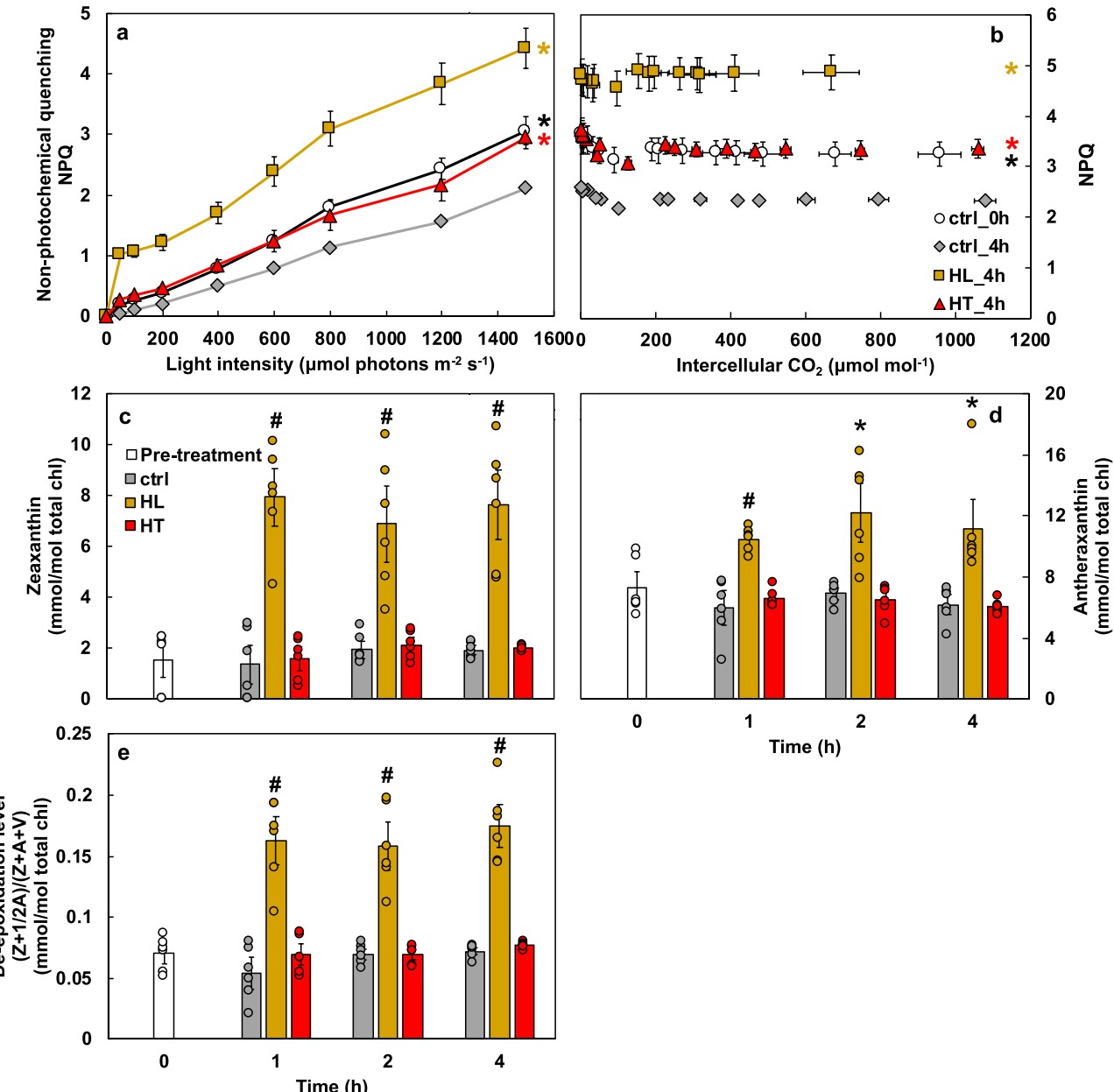

**Fig. 6 High light induced non-photochemical quenching (NPQ) and increased zeaxanthin as well as de-epoxidation levels. a** Light and **b** $CO_2$ response of NPQ. Mean ± SE, $n = 3-6$ biological replicates. Most data points of ctrl_0h, HL_4h, and HT_4h were statistically significantly different compared to ctrl_4h using Student's two-tailed $t$-test with unequal variance, denoted by asterisks at the end of curves. $p$-Values were corrected for multiple comparisons using FDR (*$p < 0.05$, the colors of * match the significance of the indicated conditions, black for ctrl_0h, yellow for HL_4h, red for HT_4h). **c–e** Concentrations of zeaxanthin, antheraxanthin, and xanthophyll cycle de-epoxidation. Mean ± SE, $n = 3$ biological replicates. Asterisk and pound symbols indicate statistically significant differences of high light or high temperature treatments compared to the control condition at the same time points using Student's two-tailed $t$-test with unequal variance (*$0.01 < p < 0.05$, #$p < 0.01$).

high-light-treated leaves. The MultispeQ $NPQ_{(T)}$ data showed that the high-light-induced NPQ was more sensitive to $pmf$ than ctrl_4h leaves, with higher NPQ produced at a given level of proton motive force in HL_4h leaves than in ctrl_4h leaves (Fig. 9d).

## Discussion

We investigated how the $C_4$ model plant *S. viridis* responds to high light or high temperature stresses at photosynthetic, transcriptomic, and ultrastructural levels (Fig. 1a) and revealed limitations of photosynthesis under these two stresses. The high light (900 μmol photons m$^{-2}$ s$^{-1}$) and high temperature (40 °C)

treatments we chose were both moderate stresses within the physiological range for *S. viridis*. Although the impact of moderate stresses can be difficult to analyze due to mild phenotypes, moderate stresses are highly relevant and occur frequently in the field[65]. Understanding the impacts of moderate stresses on $C_4$ plants is imperative for agricultural research. The moderately high light and high temperature we used reduced net $CO_2$ assimilation rates at comparable levels in *S. viridis* leaves (Fig. 1c), but via different mechanisms (Fig. 10).

**Starch over-accumulation may contribute to photoinhibition in high-light-treated leaves.** In response to high light, *S. viridis*

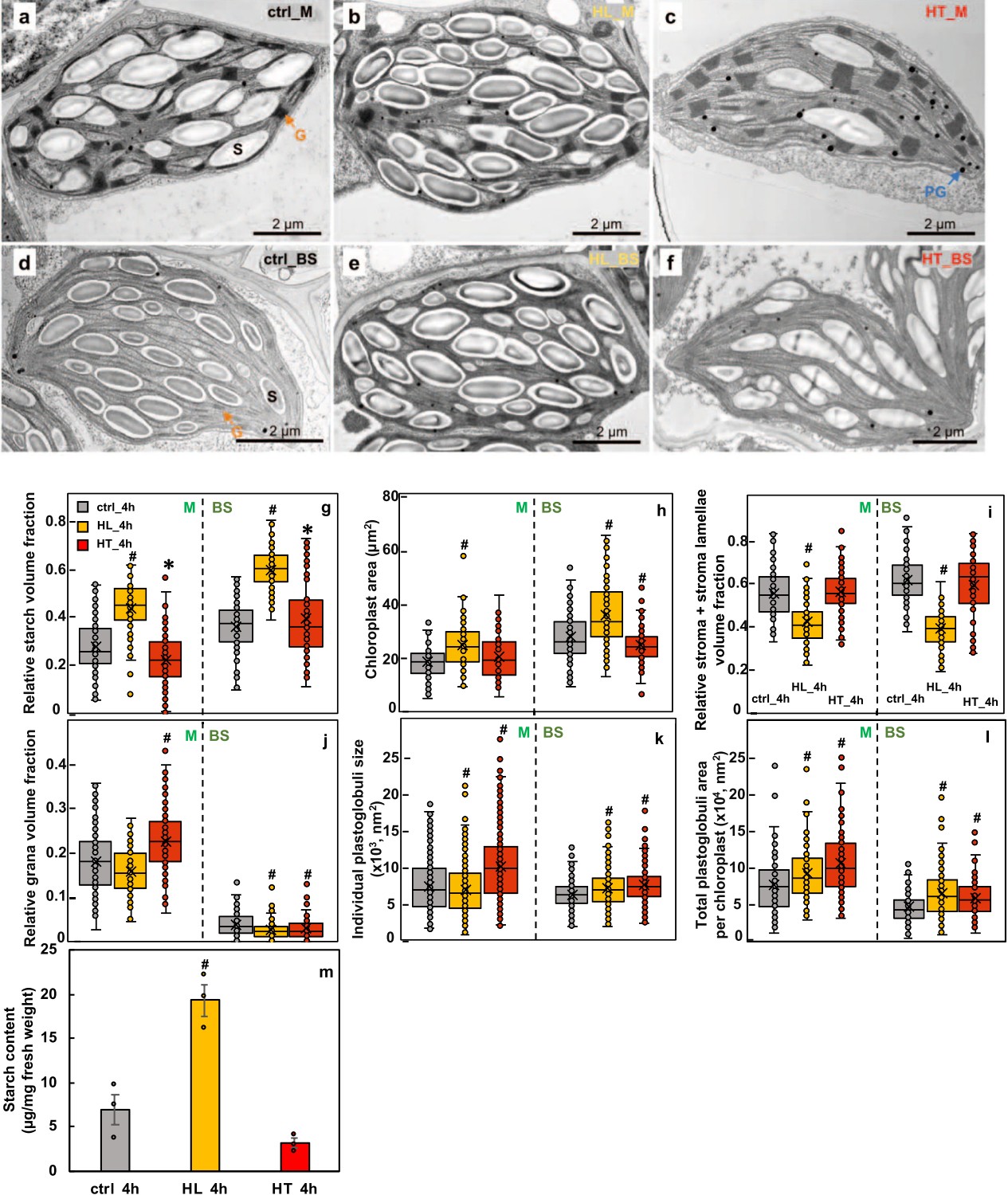

**Fig. 7 High light increased starch accumulation and both high light and high temperature treatments induced chloroplast plastoglobuli formation in *S. viridis* leaves. a–f** Representative transmission electron microscopy (TEM) images of mesophyll (M) and bundle sheath (BS) chloroplasts in leaves of *S. viridis* after 4 h treatments of control (ctrl_4h) or high light (HL_4h) or high temperature (HT_4h). TEM images of mesophyll **(a–c)** and bundle sheath **(d–f)** chloroplasts. S labels the starch granule; G labels grana, the orange arrows indicate grana in mesophyll and bundle sheath chloroplasts; PG labels plastoglobuli. **g, i, j** Relative volume fraction of indicated parameters were quantified using Stereo Analyzer with Kolmogorov–Smirnov test for statistical analysis compared to the same cell type of the control condition. **h, k, l** Area and size of indicated parameters were quantified using ImageJ with two-tailed *t*-test with unequal variance compared to the same cell type of the control condition. Each treatment had three biological replicates, total 90–120 images per treatment. *0.05 < *p* < 0.01; #*p* < 0.01. **m** Starch quantification using starch assay kits. Values are mean ± SE, *n* = 3 biological replicates. Pound symbols indicate statistically significant differences as compared to ctrl_4h using Student's two-tailed *t*-test with unequal variance (#*p* < 0.01).

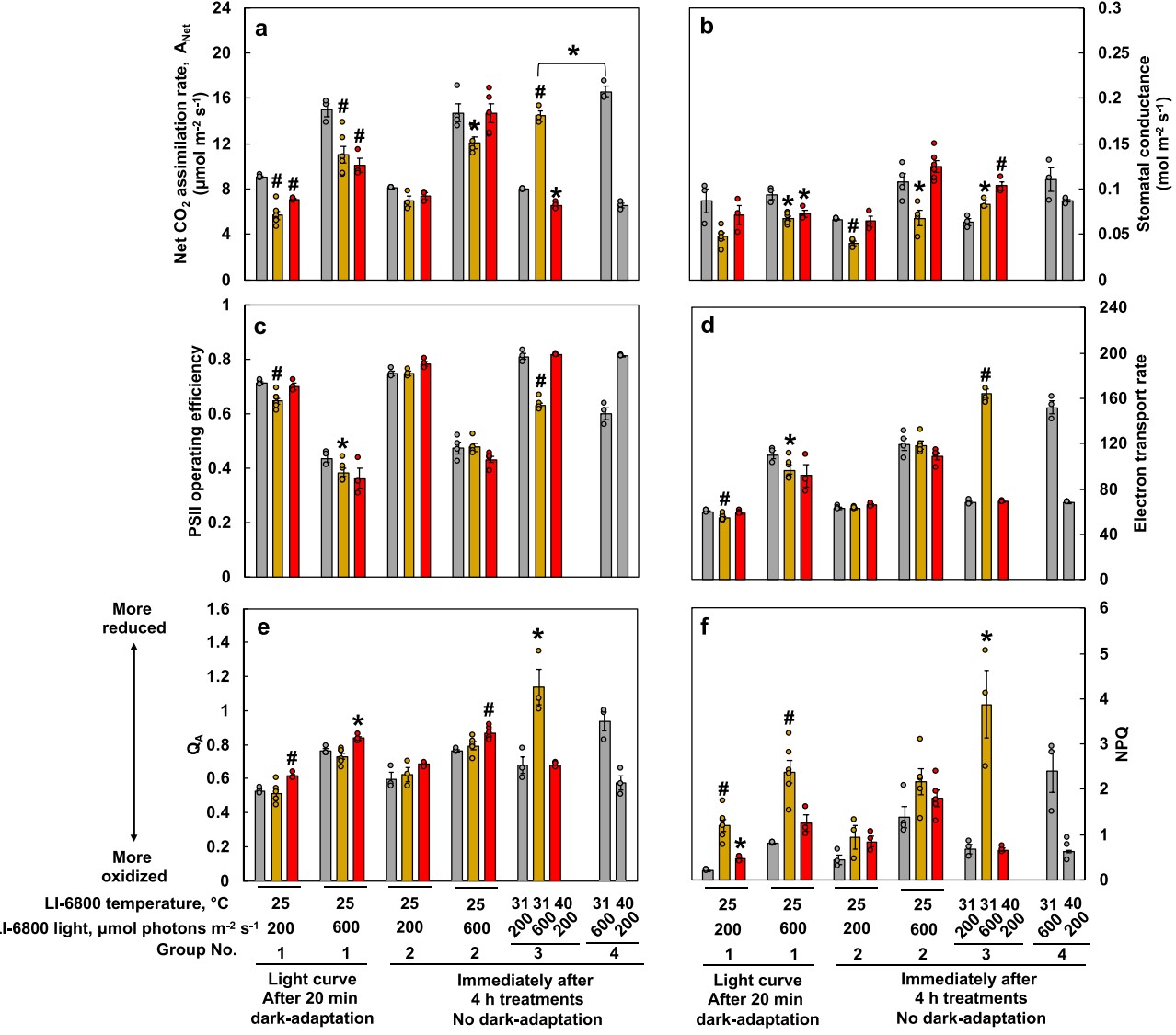

**Fig. 8 High-light- or high-temperature-treated leaves had lower photosynthetic capacities than leaves treated with the control condition.** After *S. viridis* plants were treated with 4 h of control condition (ctrl_4h) or high light (HL_4h) or high temperature (HT_4h), photosynthetic parameters in treated leaves were monitored using LI-6800. Group 1 are select data from the light response curves after 20 min dark-adaptation with indicated light and temperature. Groups 2, 3, and 4 were measured immediately after 4 h of control, high light, high temperature treatments without dark-adaptation and under the indicated temperature and light condition. **a** Net $CO_2$ assimilation rates. **b** Stomatal conductance. **c** PSII operating efficiency. **d** Electron transport rate. **e** Plastoquinone redox status ($Q_A$). **f** NPQ, Non-photochemical quenching. Values are mean ± SE, $n = 3–6$ biological replicates. Asterisk and pound symbols indicate statistically significant differences of HL_4h and HT_4h leaves compared to ctrl_4h leaves in the same group or under the same condition using Student's two-tailed *t*-test with unequal variance ($*0.01 < p < 0.05$, $#p < 0.01$).

induced NPQ to dissipate excess light energy via increased *PsbS* transcription and zeaxanthin accumulation (Figs. 3a and 6c). At the transcriptional level, high-light-treated plants up-regulated transcripts involved in PSII assembly/repair and photoprotection before down-regulating transcripts involved in LHCII, PSII core complex, and PSI complex (Fig. 3), suggesting a strategy to dissipate light and repair damaged PSII before the remodeling of photosystems. With the rapid induction of photoprotective pathways, it was initially surprising to see the significant amount of photoinhibition in high-light-treated leaves of *S. viridis* (Fig. 1b), but the high-light-induced starch accumulation may provide some insight.

Our TEM data showed that the mean relative starch volume fraction was increased significantly in both mesophyll and bundle sheath chloroplasts in HL_4h leaves as compared to ctrl_4h leaves (Fig. 7 and Supplementary Table 2). The increased starch accumulation likely resulted from increased $CO_2$ fixation rates (Fig. 8a) but imbalance of starch synthesis/ degradation and sugar transport from downstream pathways under high light. In $C_3$ plants, starch is mostly present in mesophyll chloroplasts where photosynthesis occurs[66,67]. In $C_4$ plants, starch is present in both bundle sheath and mesophyll chloroplasts (Fig. 7a–f), although Rubisco predominantly localizes in the bundle sheath chloroplasts[67]. The over-accumulated starch increased the crowdedness of the chloroplasts (Fig. 7 and Supplementary Table 2), which may hinder PSII repair, especially in mesophyll chloroplasts where PSII is enriched. PSII complexes are concentrated in the stacked grana regions; during PSII repair, damaged PSII subunits migrate from the stacked grana region to the grana margin and the unstacked grana region (stroma lamellae) where the proteins involved in PSII repair are localized (e.g., FtsH, Deg proteases that degrade

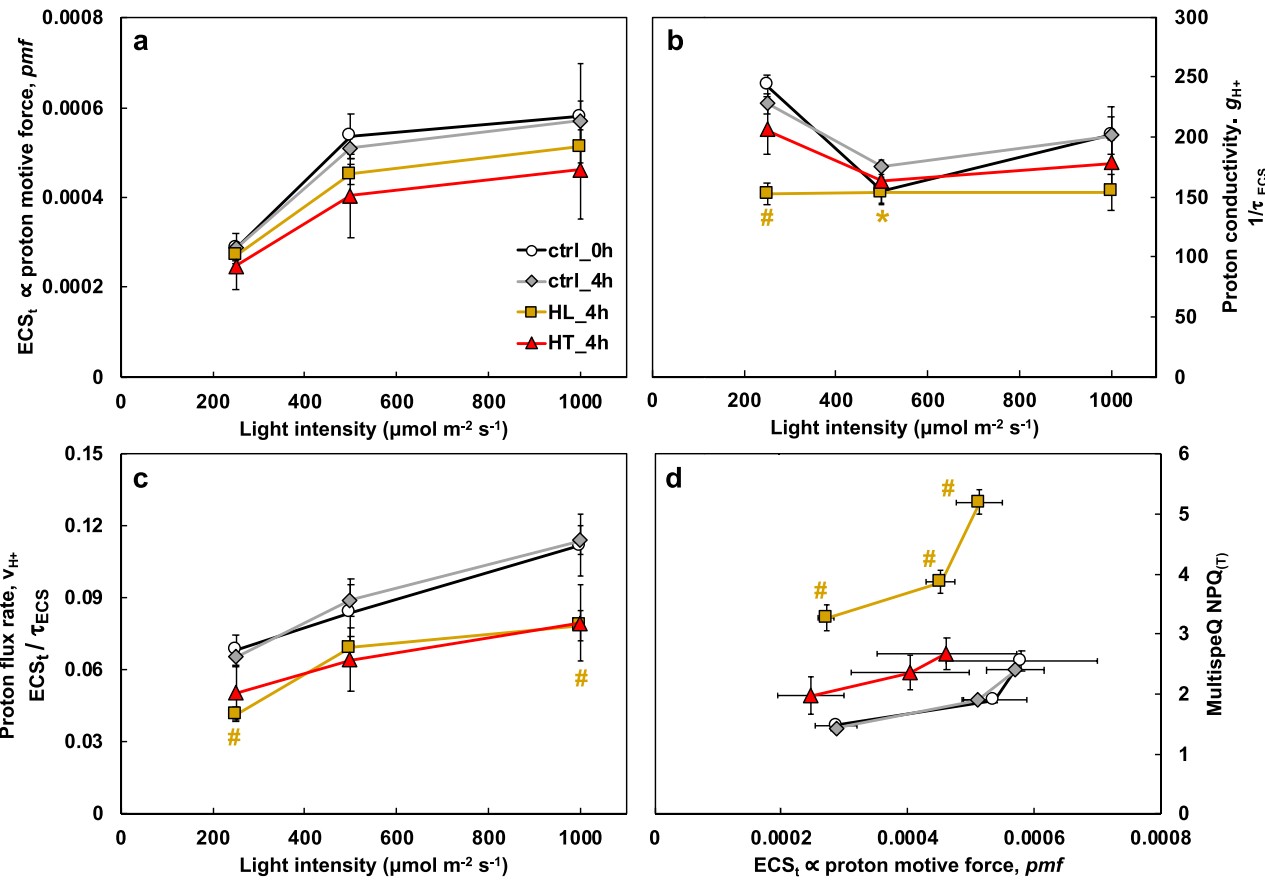

**Fig. 9 High light treatment inhibited ATP synthase activity.** After *S. viridis* plants were treated with 4 h of control condition (ctrl_4h) or high light (HL_4h) or high temperature (HT_4h), photosynthetic parameters in treated leaves were monitored using the MultispeQ instrument. **a** $ECS_t$, measured by electrochromic shift (ECS), representing the transthylakoid proton motive force, *pmf*. **b** Proton conductivity ($g_{H^+} = 1/\tau_{ECS}$), proton permeability of the thylakoid membrane and largely dependent on the activity of ATP synthase, inversely proportional to the decay time constant of light–dark-transition-induced ECS signal ($\tau_{ECS}$). **c** Proton flux rates, $v_{H^+}$, calculated by $ECS_t/\tau_{ECS}$, the initial decay rate of the ECS signal during the light–dark transition and proportional to proton efflux through ATP synthase to make ATP. **d** Non-photochemical quenching (NPQ) measured by MultispeQ. Mean ± SE, $n = 3$–6 biological replicates. Asterisk and pound symbols indicate statistically significant differences of ctrl_0h, HL_4h, and HT_4h compared to ctrl_4h using Student's two-tailed *t*-test with unequal variance. ($^*0.01 < p < 0.05$, $^\#p < 0.01$, the colors of * and # match the significance of the indicated conditions, yellow for HL_4h).

damaged PSII subunits)[15,68]. In Arabidopsis under high light, the grana lumen and margin swell to facilitate protein diffusion and PSII repair[23,69], however, we did not see these changes in high-light-treated *S. viridis* leaves (Supplementary Fig. 10d, e, i). Starch over-accumulation and increased chloroplast crowdedness may slow down the migration of damaged PSII subunits and inhibit PSII repair, contributing to the high-light-induced photoinhibition (Figs. 1b and 10). Additionally, ATP synthase activity was significantly reduced in HL_4h leaves as compared to ctrl_4h leaves (Fig. 9b, c), which may be associated with the starch accumulation and sugar feedback inhibition of photosynthesis. High-light-treated Arabidopsis plants had reduced starch in chloroplasts[70], which may reflect the differences in experimental conditions or the stronger capability to use high light for carbon fixation in $C_4$ plants than in $C_3$ plants.

**High light differentially regulated genes involved in sugar-sensing pathways**. Sugar signaling integrates sugar production with environmental cues to regulate photosynthesis[35,71,72]. In $C_3$ plants, some of the sugar-sensing pathways include: (1) SnRK1 pathway (sucrose-non-fermenting 1-related protein kinase 1, starvation sensor, active under stressful and sugar deprivation conditions to suppress growth and promote survival)[73–75]; and (2) Trehalose pathway (trehalose is a signal metabolite in plants

under abiotic stresses and helps plants survive stresses)[65,76]. In the trehalose pathway, trehalose-6-phosphate synthase (TPS) produces trehalose-6-phosphate (T6P); the T6P phosphatase (TPP) dephosphorylates T6P to generate trehalose[65]. T6P correlates with sucrose levels, inhibits SnRK1 pathway, and primes gene expression for growth in response to sucrose accumualtion[77,78]. Sugar-sensing pathways under abiotic stresses are under-explored in $C_4$ plants[35]. Research in maize showed potential inhibition of the SnRK1 pathway by T6P in reproductive tissues (e.g., kernels) under drought and salt stresses, but their roles and interaction in leaves are unclear[79,80]. Our RNA-seq data showed that two subunits of *SnRK1* ($\beta2$, $\gamma4$) were highly down-regulated under high light (Fig. 4b), suggesting possible inhibition of the SnRK1 pathway. A copy of the potential catalytically active *TPS* (*TPSI*) in *S. viridis* was induced and two copies of *TTP* were down-regulated during high light (Fig. 4b), suggesting possible increased level of T6P. Based on the expression pattern of genes involved in sugar-sensing pathways and the over-accumulated starch under high light, we postulated that high-light-treated *S. viridis* leaves had increased sugar levels, and possibly up-regulated T6P sugar-sensing pathway to down-regulate the SnRK1 pathway and promote plant growth, which may alleviate the stress of starch over-accumulation and photosynthesis inhibition under high light.

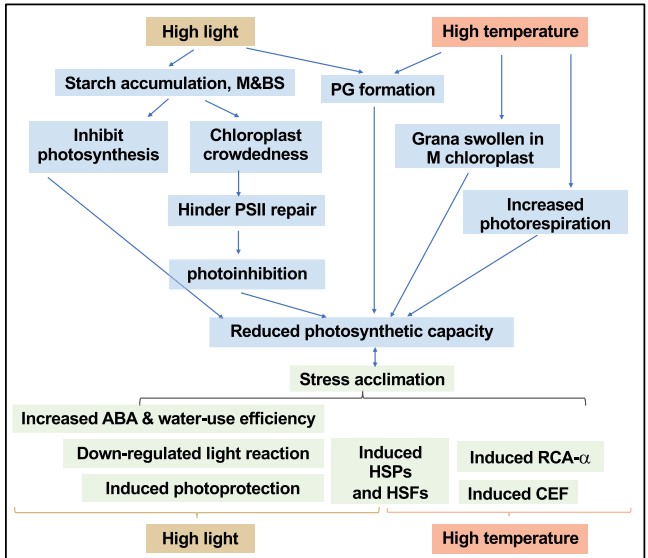

**Fig. 10 Summary of how *S. viridis* responds to high light or high temperature.** Light blue boxes denote changes that may lead to the reduced photosynthetic capacities; light green boxes denote changes that may be adaptive for high light or high temperature acclimation. M: mesophyll chloroplasts; BS, bundle sheath chloroplasts. High-light-treated leaves had over-accumulated starch and increased chloroplast crowdedness, which may hinder PSII repair and result in photoinhibition. Starch accumulation may also inhibit photosynthesis through feedback regulation. Increased plastoglobuli (PG) formation in high-light-treated leaves may affect thylakoid composition and function. Under high temperature, mesophyll chloroplasts had swollen grana and bundle sheath chloroplasts had some different responses. Heat-induced photorespiration and PG formation could further reduce photosynthesis. Meanwhile, high light and high temperature also induce adaptive responses for acclimation. Under high light, the induced photoprotection, down-regulated light reaction, and increased water-use efficiency through abscisic acid (ABA) can help *S. viridis* acclimate to excess light. Under high temperature, the induced cyclic electron flow (CEF) and Rubisco activase (RCA-α) can protect photosynthesis from heat stress. The induced heat shock transcription factors (HSFs) and heat shock proteins (HSPs) are adaptive responses to both high light and high temperature, although the induction was much quicker under high temperature than under high light.

**Potential links between high light response and ABA pathway exist in *S. viridis*.** The links between high light responses and ABA have been reported in $C_3$ plants[11,12,81,82]. Arabidopsis ABA biosynthesis mutants (e.g., *nced3*) were more sensitive to high light than WT[11,12]. High-light-treated *S. viridis* leaves had reduced capacity for stomatal conductance (Fig. 8b), which can most likely be attributed to an acute increase of ABA levels in high-light-treated leaves (Fig. 5b). Although ABA levels were only significantly increased at HL_1h and then gradually decreased, the ABA-induced stomatal closure may be prolonged. Consistent with this, RNA-seq data showed increased expression of genes involved in ABA responses and signaling during the 4 h high light treatment (Fig. 5a). Stomatal conductance increases with light to increase $CO_2$ uptake, which also increases water loss. To reduce water loss and improve water-use efficiency, a relatively lower stomatal conductance under high light may be an adaptive response. Our results in *S. viridis* provide insight into the reduced stomatal conductance and photosynthesis in sorghum leaves under high light[9].

ABA homeostasis is maintained by the balance of its biosynthesis, catabolism, reversible glycosylation, and transport pathways[19]. Several ABA biosynthesis genes were up-regulated during high light (Fig. 5a), including *NCED1* (9-*cis* epoxycarotenoid dioxygenase)[19,83,84] and *ABA1/ZEP1*, suggesting that local, de novo ABA biosynthesis may be one source of the rapid and large induction of ABA at HL_1h. The up-regulation of *CYP707As*, which are responsible for ABA degradation[85], may contribute to the gradual reduction of ABA levels after 1 h high light. Furthermore, the *S. viridis* homolog of Arabidopsis *BG1* (glucosidase, hydrolyzes inactive ABA-GE to active ABA in endoplasmic reticulum)[86] was induced at HL_2h and HL_4h. Dehydration rapidly induces polymerization of AtBG1 and a 4-fold increase in its enzymatic activity[86]. It is possible that the hydrolysis of ABA-GE to ABA by polymerized BG1 may precede the induction of the *BG1* transcript, contributing to the transiently increased ABA levels. Several putative ABA transporters were not differentially expressed (Supplementary Data 6), but a *S. viridis* homolog of the Arabidopsis ABA importer *ABCG40* was down-regulated in high light (Fig. 5a), suggesting ABA import from other parts of the plant to leaves may be less likely. Thus, the high light increased ABA level may be due to ABA de novo biosynthesis and/or reversible glycosylation from ABA-GE to ABA.

**High temperature responses had distinct features in comparison to high light.** Compared to high light, HT_4h leaves showed much less change in starch accumulation, little change in chloroplast crowdedness (Fig. 7), and no photoinhibition (Fig. 1). Grana dimension increased in high-temperature-treated mesophyll chloroplasts (Supplementary Table 2), suggesting heat-induced grana swelling. In contrast, bundle sheath chloroplasts have slightly increased starch, no change of chloroplast area, but decreased grana dimension under high temperature, suggesting cell-type-specific heat responses. PG formation was highly induced in both mesophyll and bundle sheath chloroplasts under high temperature, which may be associated with heat-increased thylakoid membrane leakiness, consistent with previous reports[26,87,88]. Induced grana swelling and PG formation may reflect heat-induced damage to chloroplast ultrastructure, which may contribute to the reduced photosynthetic rates under high temperature.

The transcriptome changes under high temperature were less extensive but more dynamic than under high light (Figs. 2–5). High temperature induced more PG formation than high light (Supplementary Table 2), however, surprisingly there were few transcriptional changes of genes encoding proteins that localize to PG under high temperature (Fig. 4a). These results suggest the heat-induced PG formation may be a direct and physical response of thylakoid membranes to moderately high temperature and not regulated at the transcriptional level.

Response to high temperature also showed some unique transcriptional changes that were absent or minimal under high light. First, high temperature resulted in high and sustained induction of Rubisco activase (*RCA-α*) (Fig. 3c). RCA removes inhibitors from Rubisco, maintains Rubisco activation, and is important for carbon fixation[48,49]. Rubisco is thermostable but RCAs are heat labile, resulting in reduced Rubisco activation and $CO_2$ fixation under HT[36]. Plants grown in warm environments usually have RCAs that are more thermotolerant[89–91]. In *S. viridis*, maize, and sorghum, high temperature induces the protein level of *RCA_α* and the rate of *RCA_α* induction is associated with the recovery rate of Rubisco activation and photosynthesis[92]. Our results, taken with previously published reports, suggest the heat-induced *RCA_α* may be the thermotolerant isoform. Understanding the function and regulation of RCAs may help improve thermotolerance of photosynthesis in $C_4$ plants. Additionally, high temperature up-regulated small HSPs much quicker than high light.

Key genes involved in photorespiration (Fig. 3c) and CEF around PSI (Supplementary Fig. 4a) were up-regulated under high temperature, suggesting high-temperature-induced photorespiration and CEF. $C_4$ plants employ carbon-concentrating mechanisms (CCM) to concentrate $CO_2$ around Rubisco and reduce photorespiration in bundle sheath chloroplasts. However, *S. viridis* bundle sheath chloroplasts have a small number of grana (Fig. 7d–f), where PSII is present and can be a source of $O_2$ production. Photorespiration increases with temperature faster than photosynthesis[30,93] and high temperature may also increase the $CO_2$ leakiness of bundle sheath chloroplasts[38,39], promoting photorespiration and reducing photosynthesis. CEF generates only ATP without NADPH, balances the ATP/NADPH ratio, generates transthylakoid proton motive force (*pmf*), and protects both PSI and PSII from photo-oxidative damage in $C_3$ plants[94,95]. Increased CEF activity has been frequently reported under various stressful conditions in $C_3$ plants[26,96,97] and in *S. viridis* under salt stress[98], indicating its important role in stress protection. To compensate for the extra ATP needed for the CCM, $C_4$ plants are proposed to have high CEF in bundle sheath chloroplasts[3,99]. Heat-induced CEF could protect photosynthesis under high temperature by maintaining transthylakoid *pmf* and generating extra ATP.

Although high light and high temperature responses had their own unique features, their transcriptional responses had significant overlaps (Fig. 2b). We identified 42 highly induced genes (FC ≥ 5) and 13 highly repressed genes (FC ≤ −5) in both conditions (Supplementary Table 1 and Supplementary Data 5). The 42 highly induced genes provide potential targets for improving resistance to high light and high temperature in $C_4$ crops, including several putative transcription factors, HSP20/70/90 family proteins, β-amylase, and a putative aquaporin transporter for promoting $CO_2$ conductivity in $C_4$ plants[3,100,101]. Additionally, *HSFA6B* was induced (2 ≤ FC ≤ 5) under both high light and high temperature. HSFA6B reportedly operates as a downstream regulator of the ABA-mediated stress response and is involved in thermotolerance in Arabidopsis, wheat, and barley[102,103]. This gene may be involved in regulation of genes that are common to both the high light and high temperature responses and it would be interesting for further study to generate high-light- and high-temperature-tolerant $C_4$ crops. Frey et al. identified 39 heat-tolerance genes in maize that were significantly associated with heat tolerance and up-regulated in most of the 8 maize inbred lines[41]. Five *S. viridis* homologs of the maize heat-tolerance genes were also up-regulated in our RNA-seq data under high temperature, providing potential engineering targets to improve heat tolerance in $C_4$ plants (Supplementary Data 5). More potential gene targets to improve high light and/or high temperature tolerance in *S. viridis* and other $C_4$ crops are included in Supplementary Data 5.

The different responses in mesophyll and bundle sheath chloroplasts in *S. viridis* are particularly interesting and warrant further study. We sorted high-light or high-temperature-induced DEGs into mesophyll- and bundle sheath-specific pathways based on previously published cell-type-specific transcriptomes[58] (Supplementary Fig. 6). Although we cannot rule out some transcripts may have altered cell-type specificity under stressful conditions, due to the functional specificity of the mesophyll and bundle sheath cells, a significant fraction of the mesophyll- and bundle sheath-specific transcripts likely keep similar cell-type specificity under our high light and high temperature conditions as compared to the published control condition. Our analysis revealed mesophyll- and bundle sheath-specific transcriptional regulation in response to high light or high temperature in *S. viridis*. Under high light, the majority of mesophyll-specific DEGs related to ROS scavenging and HSPs were up-regulated while the majority of bundle sheath-specific DEGs related to

these two pathways were down-regulated, suggesting mesophyll cells may require more ROS scavenging and HSPs than bundle sheath cells in response to high light, likely due to more ROS production and higher need for maintaining protein homeostasis in mesophyll cells than in bundle sheath cells under high light. In contrast, under high temperature, many ROS-scavenging DEGs were up-regulated in bundle sheath cells but down-regulated in mesophyll cells (possibly due to heat-induced photorespiration) while DEGs related to HSPs were up-regulated in both cell types. It is intriguing that high light up-regulated mesophyll-specific sugar transporters but down-regulated bundle sheath-specific sugar transporters. In Arabidopsis, SWEET16/17 plays a key role in facilitating bidirectional sugar transport along sugar gradient across the tonoplast of vacuoles[104,105]. The homolog of SWEET16/17 in *S. viridis* is mesophyll cell specific and was up-regulated in high light (Supplementary Fig. 4b), suggesting SvSWEET16/17 may mediate sugar uptake into vacuoles in response to a high concentration of cytosolic sugar level in mesophyll cells. The down-regulation of bundle sheath-specific SWEETs under high light may indicate feedback inhibition of phloem sugar loading due to unmatched sugar usage in downstream processes[106].

In comparison to the $C_3$ model plant Arabidopsis, the $C_4$ model plant *S. viridis* has shared and unique responses under high light and high temperature. The shared responses include induced NPQ, *PsbS* transcription, zeaxanthin accumulation, PG formation, and ABA levels under high light, and the induced PG formation under high temperature. The unique responses in *S. viridis* to high light include the over-accumulated starch in both mesophyll and bundle sheath chloroplasts and increased chloroplast crowdedness. In high temperature, the unique responses in *S. viridis* include dynamic transcriptome regulation and different heat responses of mesophyll and bundle sheath chloroplasts. Additionally, *S. viridis* has M/BS cell-type-specific transcriptional responses to high light or high temperature. The reduced photosynthetic capacity in *S. viridis* leaves under high light or high temperature also demonstrated the need to improve the tolerance to these two stresses in $C_4$ plants.

In summary, we elucidated how the $C_4$ model plant *S. viridis* responds to moderately high light or high temperature at the photosynthetic, transcriptomic, and ultrastructural levels (Supplementary Table 3). Our research furthers understanding of how $C_4$ plants respond to high light and high temperature by linking the data from multiple levels, reveals different acclimation strategies to these two stresses in $C_4$ plants, discovers unique high light/temperature responses in $C_4$ plants in comparison to $C_3$, demonstrates M/BS cell-type specificity under these two stresses, distinguishes adaptive from maladaptive responses, and identifies potential targets to improve abiotic stress tolerance in $C_4$ crops.

## Methods

**Plant growth conditions and treatments**. *S. viridis* ME034 (also known as ME034v) plants were grown in a controlled environmental chamber under constant 31 °C, 50% humidity, ambient $CO_2$ conditions, 12/12 h day/night, and leaf-level light intensity of 250 µmol photons $m^{-2} s^{-1}$. Similar level of growth light has been used frequently in literature for *S. viridis* under control conditions[58,98,107,108]. Seeds were germinated on Jolly Gardener C/V Growing Mix (BGF Supply Company, Oldcastle, OCL50050041) and fertilized with Jack's 15-5-15 (BGF Supply Company, J.R. Peters Inc., JRP77940) with an Electrical Conductivity (EC) of 1.4. At 7 days after sowing (DAS), seedlings were transplanted to 3.14″ × 3.18″ × 3.27″ pots. At 13 DAS, 4 h after light was on in the growth chamber, plants with fourth fully expanded true leaves were selected for 4 h high light (leaf-level light intensity of 900 µmol photons $m^{-2} s^{-1}$ and chamber temperature of 31 °C) or 4 h high temperature (chamber temperature of 40 °C and leaf-level light intensity of 250 µmol photons $m^{-2} s^{-1}$) treatments carried out in separate controlled environmental chambers under 50% humidity and ambient $CO_2$ conditions. A separate set of plants remained in the control growth condition.

**Gas exchange and chlorophyll fluorescence measurements**. Leaf-level gas exchange and pulsed amplitude modulated (PAM) chlorophyll $a$ fluorescence were measured using a portable gas-exchange system LI-6800 coupled with a Fluo-rometer head 6800-01 A (LI-COR Biosciences, Lincoln, NE). Fourth, fully expan-ded, intact true leaves of $S.$ $viridis$ plants from different treatments were first dark-adapted for 20 min in the LI-6800 chamber to measure maximum PSII efficiency ($F_v/F_m$) under constant $CO_2$ partial pressure of 400 ppm in the sample cell, leaf temperature 25 °C, leaf VPD 1.5 kPa, fan speed 10,000 RPM, and flow rate 500 $\mu$mol s$^{-1}$. We then performed the light response curves followed by $CO_2$ response curves ($A/c_i$ curve) as described (Supplementary Table 4). Red-blue actinic light (90%/10%) and 3–6 biological replicates for each treatment were used for all measurements. We used a leaf temperature of 25 °C for light and $CO_2$ response curves as described in previous publications for $S.$ $viridis$ regardless of growth temperatures[56,98,109–111]. During all measurements, the instrument para-meters were consistent and stable. For $CO_2$ response curves, all net $CO_2$ assim-ilation rates were corrected with the empty chamber data to account for inevitable and minor LI-6800 leaf chamber leakiness during the $CO_2$ response curves fol-lowing the established protocols[112].

Photosynthetic parameters were calculated as described[62] (see formulas, Supplementary Table 5). To estimate the true NPQ, $F_m$ used in the NPQ formula ($F_m/F_mF_m' - 1$) needs to be the maximum chlorophyll fluorescence in fully relaxed, dark-adapted leaves in which there is no quenching[62,113]. $F_m$ and $F_mF_m'$ are the maximum chlorophyll fluorescence yields in dark-adapted and light-adapted leaves, respectively[62,113,114]. In control leaves, $F_m$ could be reached with 20 min dark-adaptation without further change after that, but high-light-treated leaves needed a much longer recovery period to relax the quenching processes due to the light-induced photoinhibition (Supplementary Fig. 7a). Because the values of $F_m$ in dark-adapted ctrl_4h leaves were highly consistent among different biological replicates and reflected the reference level of $F_m$ (i.e., without stress treatments), we used the mean $F_m$ of ctrl_4h leaves as a baseline to calculate NPQ in leaves with different treatments.

To investigate photosynthetic performance in plants immediately following 4 h of different treatments (control, high light or high temperature), we also performed short LI-6800 measurements for 5 min on each plant immediately after 4 h treatments without dark-adaptation at 400 ppm $CO_2$ with indicated leaf temperatures and light intensities (Fig. 8). To estimate photosynthetic parameters under different treatments as in the growth chambers, the LI-6800 leaf chamber was set to simulate the condition of different treatments: control (31 °C, 200 $\mu$mol photons m$^{-2}$ s$^{-1}$ light), high light (31 °C, 600 $\mu$mol photons m$^{-2}$ s$^{-1}$ light,) or high temperature (40 °C, 200 $\mu$mol photons m$^{-2}$ s$^{-1}$ light). The temperature and light refer to the conditions in the LI-6800 leaf chamber. The light in LI-6800 leaf chamber (90% red and 10% blue) was different from the white light in our growth chambers, therefore we selected two lights (200 and 600 $\mu$mol photons m$^{-2}$ s$^{-1}$) in the LI-6800 leaf chamber that were close to the white lights in growth chambers based on the light quantification in the red (580–670 nm) and blue (440-540 nm) spectrum range. LI-6800 light intensities of 200 and 600 $\mu$mol photons m$^{-2}$ s$^{-1}$ were also two of the conditions used in the light response curves with dark-adaptation (Figs. 1c and 8, group 1), allowing for direct comparison. Individual plants were used for each replicate.

The high abundance of PSI in bundle sheath chloroplasts of C$_4$ leaves can affect chlorophyll fluorescence measurement (up to 50%) and underestimate the PSII efficiency ($F_v/F_m$) and electron transport rates[115,116]. Thus, our chlorophyll fluorescence data were corrected with 0.5 $F_o$[116]. $F_o$ is the mean minimal chlorophyll fluorescence in dark-adapted leaves under the control condition (ctrl_4h). The PSII operating efficiency calculated from the corrected and uncorrected chlorophyll fluorescence data correlated with each other but the corrected data yielded higher PSII efficiency, with the maximum PSII efficiency in ctrl_4h leaves closer to the theoretical values of 0.86[117] (Fig. 1b).

**Modeling of photosynthetic parameters using leaf-level gas-exchange infor-mation**. To model photosynthetic parameters, we used gas-exchange data from light response curves and $CO_2$ response curves ($A/c_i$ curves). The model para-meterization and analyses were conducted in R 3.4.3 Project software® (R Devel-opment Core Team 2016). First, light response curves were fitted as previously described[118]. We fit a non-linear least squares regression (non-rectangular hyperbola) to estimate photosynthetic parameters (Supplementary Fig. 3). $A/c_i$ curves were fitted as previously described[119] to estimate the $V_{cmax}$ (the maximum rate of carboxylation). We used the C$_4$ photosynthesis model using a Bayesian analysis approach as described in Feng et al. (2013)[120]. The normality of the data was verified with the Shapiro-Wilk test. Statistical analysis was performed using Student's two-tailed $t$-test with unequal variance by comparing ctrl_4h with all other conditions.

**RNA isolation**. To isolate RNA from leaves, four biological replicates containing two 2-cm middle leaf segments from two plants for each time point and treatment were collected from fourth fully expanded true leaves into screw cap tubes (USA Scientific, 1420-9700) with a grinding bead (Advanced Materials, 4039GM-S050) and immediately frozen in liquid nitrogen and stored at −80 °C. Frozen samples were homogenized using a paint shaker. RNA was extracted using a Trizol method with all centrifugation at 4 °C and 11,000 RCF. First, 1 mL of Trizol Reagent

(Invitrogen, 15596018) was added to homogenized leaf tissue and resuspended, then 200 $\mu$L of Chloroform:Isoamyl alcohol (25:1) was added and vortexed. Tubes were centrifuged for 15 min, and 600 $\mu$L from the aqueous layer was transferred to a clean tube with equal volume Chloroform:Isoamyl alcohol, vortexed, and cen-trifuged for 5 min. Next, 450 $\mu$L of aqueous layer was transferred to 0.7x volume 100% Isopropanol, mixed well, and chilled for 30 min in −20 °C freezer. Samples were centrifuged for 15 min to pellet RNA. Supernatant was decanted, and RNA pellet was rinsed twice with ice-cold 75% ethanol with a 2 min centrifugation following each rinse. RNA was dried in a laminar flow hood until residual ethanol evaporated and was resuspended in 50 $\mu$L of nuclease-free H$_2$O. RNA was quan-tified using a NanoDrop and Qubit RNA Broad Range (BR) Assay Kit (Thermo Fisher Scientific Inc., Q10210) with the Qubit 3.0 machine. RNA integrity was verified using a Bioanalyzer Nano Assay (Genome Technology Access Center, Washington University in St Louis).

**RNA-seq library construction and sequencing**. RNA samples were diluted to 200 ng/$\mu$L in nuclease-free H$_2$O for a total of 1 $\mu$g RNA. Libraries were generated with the Quantseq 3′ mRNA-seq library prep kit FWD for Illumina (Lexogen, 015.96). Libraries were generated according to manufacturer's instructions. Cycle count for library amplification for 1 $\mu$g mRNA was tested using the PCR add-on kit for Illumina (Lexogen, 020.96). qPCR was performed and a cycle count of 13 was determined for the amplification of all libraries. For library amplification, the Lexogen i5 6 nt Dual Indexing Add-on Kit (5001-5004) (Lexogen, 047.4 × 96) was used in addition to the standard kit to allow all libraries to have a unique com-bination of i5 and i7 indices. All libraries were quantified using Qubit dsDNA High Sensitivity (HS) Assay Kit (Thermo Fisher Scientific Inc., Q32854) with the Qubit 3.0 machine. Prepared libraries were pooled to equimolar concentrations based on Qubit assay reads. Pooled libraries were submitted to Novogene to be sequenced on the HiSeq4000 platform (Illumina) with paired end, 150 bp reads.

**Mapping and transcript quantification**. Single-end reads were trimmed and quality-checked using Trim Galore (version 0.6.2). Trimmed reads from each library were mapped and processed for transcript quantification using Salmon (version 1.1.0) in quasi-mapping mode with a transcriptome index built from the $S.$ $viridis$ transcript and genome files (Sviridis_311_v2; Phytozome v12.1, sequence data produced by the US Department of Energy Joint Genome Institute and the $S.$ $viridis$ Genome Sequencing Project)[42]. Salmon outputs were imported into R using the Bioconductor package tximport (1.16.0) to extract gene-level expression values represented by transcript per million (TPM) for each gene across every time point, tissue, and treatment group sampled. PCA was performed with TPM normalized read counts of all genes using the R package FactoMineR[121].

**Differential expression analysis**. Genes that met minimum read count cutoffs of at least 10 raw reads in at least 10% of samples (14,302 genes) were included in differential expression analysis using DeSeq2, FDR < 0.05[122]. High light or high temperature treatment time points were compared to the control condition from the same time point. DEGs between different time points in either high light or high temperature were visualized in UpSetR[123]. To identify genes in key pathways of interest in $S.$ $viridis$, we used the MapMan annotations for the closely related $S.$ $italica$ (RRID:SCR_003543). From the $S.$ $italica$ MapMan annotations, we identified the best hit in $S.$ $viridis$ for genes in pathways of interest. We then manually curated these lists based on relevant literature to obtain genes in pathways of interest (Supplementary Data 6), as well as to provide further annotation information for genes identified using the MapMan annotations. We sorted the differentially expressed genes in pathways of interest into fold change (FC) bins based on their DeSeq2 fold change values and presented their expression patterns. FC bins were defined as follows: highly induced: FC ≥ 5; moderately induced: 5 > FC ≥ 2; slightly induced: 2 > FC > 0; not differentially expressed: FC = 0; slightly repressed: 0 > FC > −2; moderately repressed: −2 ≥ FC > −5; highly repressed: FC ≤ −5. Heatmaps of pathways of interest were generated using the R package pheatmap (version 1.0.12. https://CRAN.R-project.org/package=pheatmap).

**ABA quantification**. Leaf samples of three biological replicates were harvested at 0, 1, 2, and 4 h of control, high light, or high temperature treatment. The fresh leaf weight was immediately measured after harvesting. The samples were quickly placed in liquid nitrogen and then stored in −80 °C freezer until further processing. Frozen leaf tissue was homogenized and 15 ng of [$^2$H$_6$]-abscisic acid was added as an internal standard. Samples were dried to completeness under vacuum. ABA was resuspended in 200 $\mu$L of 2% acetic acid in water (v/v) and then centrifuged; an aliquot was then taken for quantification. Foliar ABA levels were quantified by liquid chromatography tandem mass spectrometry with an added internal standard using an Agilent 6400 Series Triple Quadrupole liquid chromatograph associated with a tandem mass spectrometer according to the previously described methods[124].

**Pigment analysis**. Three biological replicates of one 2 cm middle leaf segment were collected from fourth fully expanded true leaves into screw cap tubes (USA Scientific, 1420-9700) with a grinding bead (Advanced Materials, 4039GM-S050), immediately frozen in liquid nitrogen, and stored at −80 °C. During pigment

extraction, 600 μL ice-cold acetone were added to the samples before they were homogenized in a FastPrep-24 5G (MP Biomedicals) at 6.5 m s$^{-1}$ for 30 s at room temperature. Cell debris were removed by centrifugation at 21,000$g$ for 1 min. The supernatant was filtered through a 4 mm nylon glass syringe prefilter with 0.45 μm pore size (Thermo Scientific) and analyzed by HPLC. HPLC analyses were performed on an Agilent 1100 separation module equipped with a G1315B diode array and a G1231A fluorescence detector; data were collected and analyzed using Agilent LC Open Lab ChemStation software. Pigment extracts were separated on a ProntoSIL 200-5 C30, 5.0 μm, 250 mm × 4.6 mm column equipped with a ProntoSIL 200-5-C30, 5.0 μm, 20 mm × 4.0 mm guard column (Bischoff Analysentechnik) and gradient conditions as previously described[125]. Assuming interconversion of the intermediate antheraxanthin between both zeaxanthin and violaxanthin, the de-epoxidation level can be calculated by (zeaxanthin + 0.5 antheraxanthin) / (violaxanthin + antheraxanthin + zeaxanthin)[126].

**Transmission electron microscopy (TEM)**. *S. viridis* leaves were collected after 4 h of different treatments and prepared for TEM. Four-millimeter biopsy punches were taken from the middle leaf segments of the fourth fully expanded leaves and fixed for 2 h in 2% paraformaldehyde and 2% glutaraldehyde (EM Science, Hatfield, PA, USA) plus 0.1% Tween20 in 0.1 M sodium cacodylate at pH 7.4 at room temperature and then at 4 °C overnight. Samples were then rinsed 3x in buffer and fixed in 2% osmium tetroxide (EM Science, Hatfield, PA, USA) in ELGA water for 2 h, rinsed 3x in ELGA water and placed in 1% uranyl acetate in ELGA water at 4 °C overnight and then at 50 °C for 2 h. Next, samples were rinsed 5x in water, dehydrated in a graded acetone series and embedded in Epon-Araldite (Embed 812, EM Science, Hatfield, PA, USA). Embedments were trimmed and mounted in the vise-chuck of a Leica Ultracut UCT ultramicrotome (Leica, Buffalo Grove, IL, USA). Ultrathin sections (~60 to 70 nm) were cut using a diamond knife (type ultra 35 °C; Diatome), mounted on copper grids (FCFT300-CU-50, VWR, Radnor, PA, USA), and counterstained with lead citrate for 8 min[25]. Samples were imaged with a LEO 912 AB Energy Filter Transmission Electron Microscope (Zeiss, Oberkochen, Germany). Micrographs were acquired with iTEM software (ver. 5.2) (Olympus Soft Imaging Solutions GmbH, Germany) with a TRS 2048 × 2048k slow-scan charge-coupled device (CCD) camera (TRÖNDLE Restlichtverstärkersysteme, Germany). Ninety electron micrographs were quantified for each experimental treatment using image analysis (FIJI software, National Institutes of Health) and stereology (Stereology Analyzer version 4.3.3, ADCIS, France). Each TEM image was acquired at 8,000X magnification and 1.37 nm pixel resolution with arrays of up to 5 × 5 tiles using automated Multiple Image Alignment software module (settings: correlation = 1, FFT algorithm, overlap area = linear weighted, movement = emphasize, and equalize). TEM images were analyzed with Stereology Analyzer software version 4.3.3 to quantify relative volume of various cell parameters including stroma, stroma lamellae, starch granules, and grana within individual chloroplasts (Supplementary Fig. 9b). Grid type was set as "point" with a sampling step of 500 × 500 pixels and pattern size of 15 × 15 pixels. The percent of relative volume for each parameter was collected after identifying all grid points within one chloroplast and further analyzed in excel. TEM images with a magnification of 8 K were used in the Fiji (ImageJ) analysis. The images were scaled to 0.7299 pixel nm$^{-1}$ in ImageJ before analyzing the chloroplast area, plastoglobuli area, and grana dimensions. The height of grana margin (positions 1 and 3) and grana core (position 2) were quantified as described previously[23] (Supplementary Fig. 10d, e). The "polygon selections" tool was used to quantify the chloroplast and plastoglobuli area by outlining the target structure. The individual plastoglobuli (PG) size was measured using ImageJ. All PG in a chloroplast were quantified to get the total PG area per chloroplast. The "straight" tool was used to quantify grana height and width. The grana number and PG number were counted manually. Choosing the correct statistical test to reflect the quantified data is essential in making conclusions. Three different statistical tests were used to find the significance of p-values. The negative binomial test was used for counting data that followed a negative binomial distribution. The Kolmogorov-Smirnov test was used for relative volume data since it is commonly used to find significance between data in a form of ratios. A two-tailed t-test with unequal variance was used for all other data that followed a normal distribution. All three statistical tests compared the treatment conditions to the control conditions of the same cell type. Each treatment had three biological replicates and a total of 90~120 images of each treatment were analyzed.

**Starch quantification**. To isolate starch from leaves, three biological replicates of 2 cm middle leaf segments were collected from fourth fully expanded true leaves into screw cap tubes (USA Scientific, 1420-9700) with a grinding bead (Advanced Materials, 4039GM-S050) and immediately frozen in liquid nitrogen and stored at −80 °C. Frozen samples were homogenized using a paint shaker. For starch quantification, leaves decolorized by 80% ethanol and starch concentration was subsequently measured using a starch assay kit (Megazyme, K-TSTA-100A).

**MultispeQ measurement**. A MultispeQ[61] v2.0 was used to measure chlorophyll fluorescence parameters and ECS in *S. viridis* leaves at the start or after 4 h treatments of control, high light, or high temperature. ECS results from light–dark-transition-induced electric field effects on carotenoid absorbance bands[62,127] and is

a useful tool to monitor proton fluxes and the transthylakoid proton motive force (*pmf*) in vivo[63,64]. Light drives photosynthetic electron transport along the thylakoid membrane and proton fluxes across the thylakoid membrane. Protons flux into the thylakoid through $H_2O$ oxidation at PSII and plastoquinol oxidation at cytochrome $b_6f$ complex; protons flux out of the thylakoid mainly through ATP synthase to make ATP, which is driven by the transthylakoid *pmf*[63,64]. The total amplitude of ECS signal during the light–dark transition, $ECS_t$, represents the transthylakoid *pmf*. The decay time constant of light–dark-transition-induced ECS signal, $\tau_{ECS}$, is inversely proportional to proton conductivity ($g_H^+ = 1/\tau_{ECS}$), which is proportional to the aggregate conductivity (or permeability) of the thylakoid membrane to protons and largely dependent on the activity of ATP synthase[62]. The proton flux rates, $v_{H+}$, calculated by $ECS_t/\tau_{ECS}$, is the initial decay rate of the ECS signal during the light–dark transition, and reflects the rate of proton translocation by the entire electron transfer chain, usually predominantly through the ATP synthase[62]. ECS was measured using MultispeQ and the dark interval relaxation kinetics with a modified Photosynthesis RIDES protocol at light intensities of 250, 500, and 1000 μmol photons m$^{-2}$ s$^{-1}$. The MultispeQ v2.0 was modified with a light guide mask to improve measurements on smaller leaves. Parameters at the different light intensities were measured sequentially on the middle segment of a fourth fully expanded true leaf at room temperature with no dark adaptation prior to measurements. The estimated NPQ, $NPQ_{(T)}$, was measured by MultispeQ based on a method that does not require a dark-adapted state of the leaf for determination of $F_m$[59]. $NPQ_{(T)}$ uses the minimal fluorescence ($F_o'$) and maximal fluorescence ($F_m'$) in light-adapted leaves to estimate NPQ[59]. Statistical significance was assigned with a two-tailed t-test assuming unequal variance.

**Statistics and reproducibility**. All data presented had at least three biological replicates. Detailed information about statistical analysis were described for each method above.

**Reporting summary**. Further information on research design is available in the Nature Research Reporting Summary linked to this article.

## Data availability
The datasets analyzed in this paper are included in this published article and supplementary information files. The RNA-seq data discussed in this publication was deposited in NCBI's Gene Expression Omnibus (GEO)[128] and are accessible through GEO Series accession number GSE178320. Other information is available from the corresponding author on reasonable request.

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

## Acknowledgements

The research was supported by the Defense Advanced Research Projects Agency (DARPA) (HR001118C0137 to R.Z., A.E., D.N., and R.V.) and start-up funding from Donald Danforth Plant Science Center (DDPSC, to R.Z.) E.M. was supported by the William H. Danforth Fellowship in Plant Sciences and Washington University in St Louis. D.P. was supported by the Berkeley Fellowship and the NSF Graduate Research Fellowship Program Grant DGE 1752814. O.D. was supported by the Deutsche Forschungsgemeinschaft (DFG) - Project number 427925948. K.K.N. is an investigator of the Howard Hughes Medical Institute. We would like to thank Dr. Helmut Kirchhoff and Dr. Charles Pignon for helpful discussion about the TEM data and LI-6800 data, respectively. We also want to thank Drs. Blake Meyers, Sona Pandey, and Ivan Baxter for their valuable feedback about the manuscript. We acknowledge imaging support from the Advanced Bioimaging Laboratory (RRID:SCR_018951) at DDPSC and usage of the LEO 912AB Energy Filter TEM acquired through a National Science Foundation (NSF) Major Research Instrumentation grant (DBI-0116650). We also acknowledge the use of the Metabolite Profiling facility of the Bindley Bioscience Center, a core facility of the National Institute of Health-funded Indiana Clinical and Translational Sciences Institute, for assisting in the quantification of ABA levels. We appreciate the DDPSC Plant Growth Facilities for growth chamber reservation and taking care of our plants. Additionally, we thank the US Department of Energy Joint Genome Institute and the *S. viridis* Genome Sequencing Project for the *S. viridis* v2.1 genome database. DARPA approved the paper for Public Release, Distribution Unlimited.

## Author contributions

R.Z. supervised the whole project. R.Z. and C.M.A. designed and planned all the experiments. C.M.A. led the project, performed and analyzed all LI-6800 data, extracted RNA and prepared the RNA-seq library, and led sample harvest for pigment analysis. C.M.A. and E.B. grew all plants needed for the project. T.J.A. provided insight for optimized gas exchange and chlorophyll fluorescence measurements in *S. viridis* using the LI-6800. T.T. and R.V. performed modeling of the leaf-level gas-exchange data. N.Z., E.B., and K.J.C. performed TEM analysis. N.Z. quantified starch using assay kits. W.M. helped harvest leaf tissues and performed MultispeQ measurements. E.B. harvested leaves for ABA measurements and S.A.M.M. performed ABA analyses. D.P., O.D., and K.K.N. performed leaf pigment analysis by HPLC. M.B. and A.L.E. provided insight for RNA-seq library preparation. J.Y. and A.L.E. preprocessed RNA-seq data. E.M. led RNA-seq data analysis and generated all the heatmaps. S.P. and R.Z. identified ABA-related genes in *S. viridis*. E.M. identified all other genes used for the heatmaps. J.B. provided suggestions for statistical analysis. M.W. and D.A.N. helped plan, coordinate, and discuss the RNA-seq experiments. R.Z., C.M.A., and E.M. led the writing of the manuscript with the contribution of all other co-authors. All the authors discussed the results, contributed to data interpretation, and helped revise the manuscript.

## Competing interests

The authors declare no competing interests.
