## [Peer Review File. · Communications Biology]

Reviewers' Comments:

Reviewer #1:

Remarks to the Author:

The paper compares the response of the C4 model to short term treatments of high light or high temperature. Overall this is a carefully conducted study with clear outcomes. The outcomes themselves are not especially surprising or novel e.g. lots of photosynthesis related genes affected by high light; heat shock proteins affected by high temperature. In general I think the writing is overly long in all sections and could be condensed.

Identification of targets and the end of the Abstract is an important element of the study, but please identify what you think are the promising targets to emerge. This will make the paper more interesting and likely enable it to pick up more citations.

Another aspect which is not adequately considered or represented is trehalose metabolism and SnRK1 which are affected by the treatments and particularly under high light. The effects are as strong as those for ABA. However, discussion of trehalose metabolism is not sufficient or accurate. There are several papers that look at expression of T6P and SnRK1 related genes in response to environmental treatments in C4 maize which are relevant for this study e.g. papers by C Henry, S Bledsoe and M Oszvald as first authors of recent papers in *Plant Physiology* and *BMC Plant Biology*. Citation 79 is about AGPase, not about SnRK1 and growth. It is important to cite papers accurately and better the specific primary articles rather than general reviews. A relevant citation about T6P, SnRK1 and growth is C Nunes et al. 2013, *Plant Physiology*.

Line 435 TTP should be TPP.

Reviewer #2:

Remarks to the Author:

The paper claims that they investigated the short term (after 4 h) effect of HL and HT on a C4 model plant. It also claims that they discover unique light/ temperature respond in C4 plants compare to C3.

Their results can identify potential targets to improve stress tolerance in C4 plants. The work has studied several pathways involved in plant stress respond, which can be the interest of other community. The paper presents a huge amount of work. However, the results are mostly coming from physiological studies. I think enzyme assays or/and metabolic studies are necessary to draw a steadier conclusion on what is the regulation of these pathways in this stress condition in the plants. But this can be a future work. Using moderate stress is a good approach as it is mimicking what can happen in the fields.

The statistical analysis of the work is valid, the materials and methods are well detailed, giving the ability to reproduce the work. The paper is well written and easy to follow.

I found some of the graph (Figure 1) a bit crowded with the label of different significant level.

Reviewer #3:

Remarks to the Author:

I have reviewed the article entitled "High Light and High Temperature Reduce Photosynthesis via Different Mechanisms in the C4 Model *Setaria viridis*". Plants were grown at 250 PPFD of light at 31°C in growth chamber. On 13 days after sowing, one set of plants were placed under 900 PPFD and 31°C for 4 hours and that was called the high light treatment, and another set of plants were placed at 250 PPFD and 40°C for 4 hours and that was called the high temperature treatment. After the treatment, gas exchange and leaf chlorophyll fluorescence were measured. Leaves also were sampled for transmission electron microscopy images, transcriptome and abscisic acid analysis. The experiment was poorly designed and does not reproduce a real problem. First: high light is only a problem to C4 plants at low temperature. At high temperature like 31°C, near the thermal optimum for most of the C4 plants, photosynthesis is believed to be limited by the ETR. In fact, C4 plants are noted to have a higher quantum yield than C3 plants. Second, light levels of 250 PPFD are really low to grow C4 plants. The high light treatment of 900 PPFD is still pretty low light. In regions/seasons where temperature reaches 31°C, light level at field condition frequently overcomes 1300 PPFD. And this is actually what the data shows: that photosynthesis is limited by

light! Whenever light availability was greater, photosynthesis boosted up. Thus, light levels of 250-900 PPFD are very unnatural and does not reflect what we find in nature. I do believe that there are some gene expression related to high light stress. But I believe that a lot of genes that would naturally express in field conditions did not express in growth chamber conditions and might have been expressed after the plant finally reached light levels near field condition. Which genes are naturally expressed in the field and which genes will just express under high light stress? Figures were also poorly designed, frequently too busy, and hard to understand. Legends are lengthy with a lot of unnecessary text. Figure 7 for example should be two different figures.

We thank the reviewers for reading our manuscript and providing valuable comments. We revised the manuscript according to the suggestions from the reviewers. Please see our responses to each of the comments in purple below. We also tracked the modifications in the text of the revised manuscript.

Frequently Used Abbreviations: *S. viridis*, *Setaria viridis*; HL, high light; HT, high temperature; M, mesophyll; BS, bundle sheath; TEM, transmission electron microscopy.

Reviewers' comments:

Reviewer #1 (Remarks to the Author):

“The paper compares the response of the C₄ model to short term treatments of high light or high temperature. Overall this is a carefully conducted study with clear outcomes.”

We thank the reviewer for the encouraging comment.

“The outcomes themselves are not especially surprising or novel e.g. lots of photosynthesis related genes affected by high light; heat shock proteins affected by high temperature.”

The transcriptional changes the reviewer mentioned are well-studied in C₃ plants, but much less so in C₄ plants. Unlike C₃ plants, studies on how C₄ plants respond to HL or HT are largely limited, especially the underlying coordination between M and BS cells and the multi-level effects of HL and HT on photosynthesis, transcriptomes, and ultrastructure of C₄ plants.

In this study, we provided a systematic analysis of HL and HT responses in the C₄ model plant *S. viridis* to help fill this knowledge gap. Besides the transcriptional changes, the novelty of our research also included the M/BS specific responses to HL and HT based on our TEM and transcriptome analysis. From our TEM analysis, we also reported the potential effects of starch over-accumulation on chloroplast ultrastructures in M and BS cells of C₄ plants.

Additionally, we showed the transient increase of ABA levels under HL which may contribute to the reduced capacity of stomatal conductance in HL-treated *S. viridis* leaves. ABA has been linked with HL response in C₃ plants, but our research is probably the first such report in C₄ plants, with supporting evidence from our data of RNA-seq, gas exchange, and ABA quantification.

Furthermore, we compared the HL and HT responses in *S. viridis* and revealed that *S. viridis* employed different acclimation strategies to HL and HT although these two treatments resulted in comparable reduction of photosynthesis. Our research will help improve the understanding of C₄ photosynthesis under abiotic stresses, especially HL and HT.

“In general I think the writing is overly long in all sections and could be condensed.”

We tried our best to trim the manuscript without losing essential information. Our manuscript contains a large amount of useful experimental data about how the C₄ model

plant *S. viridis* responds to HL or HT at multiple levels. We also followed the journal guideline for clarity with the necessary details.

“Identification of targets and the end of the Abstract is an important element of the study, but please identify what you think are the promising targets to emerge. This will make the paper more interesting and likely enable it to pick up more citations.”

We thank the reviewer for the suggestion. We briefly mentioned some potential gene targets in the Discussion (previous version, line 533-544). We now modified Supplementary Data 5 to emphasize the potential gene targets that may help *S. viridis* and other C₄ plants tolerate HL and/or HT. We also modified the legend of Supplementary Data 5 to reflect this change.

“Another aspect which is not adequately considered or represented is trehalose metabolism and SnRK1 which are affected by the treatments and particularly under high light. The effects are as strong as those for ABA. However, discussion of trehalose metabolism is not sufficient or accurate. There are several papers that look at expression of T6P and SnRK1 related genes in response to environmental treatments in C₄ maize which are relevant for this study e.g. papers by C Henry, SBledsoe and M Oszvald as first authors of recent papers in Plant Physiology and BMC Plant Biology.”

We thank the reviewer for the suggestions. We cited the papers the reviewer mentioned and modified the discussion about the trehalose metabolism and SnRK1.

“Citation 79 is about AGPase, not about SnRK1 and growth. It is important to cite papers accurately and better the specific primary articles rather than general reviews. A relevant citation about T6P, SnRK1 and growth is CNunes et al. 2013, Plant Physiology.”

We thank the reviewer for the suggestion. We corrected the citation and also cited the paper the reviewer mentioned.

Nunes, C. et al. The trehalose 6-phosphate/SnRK1 signaling pathway primes growth recovery following relief of sink limitation. *Plant Physiology* 162, 1720–1732 (2013).

Reviewer #2 (Remarks to the Author):

“The paper claims that they investigated the short term (after 4 h) effect of HL and HT on a C₄ model plant. It also claims that they discover unique light/ temperature respond in C₄ plants compare to C₃. Their results can identify potential targets to improve stress tolerance in C₄ plants. The work has studied several pathways involved in plant stress respond, which can be the interest of other community. The paper presents a huge amount of work.”

We thank the reviewer for the recognition of the effort and novelty of our study.

“However, the results are mostly coming from physiological studies. I think enzyme assays or/and metabolic studies are necessary to draw a steadier conclusion on what is the regulation of these pathways in this stress condition in the plants. But this can be a future work.”

Our results included data from multiple levels, including photosynthetic measurements, transcriptome, chloroplast ultrastructural changes using TEM images, and ABA and pigment quantification. We agree with the reviewer that enzyme assays or/and metabolic studies can be helpful to understand more about how C₄ plants respond to HL or HT. We will consider these experiments for future research.

“Using moderate stress is a good approach as it is mimicking what can happen in the fields. The statistical analysis of the work is valid, the materials and methods are well detailed, giving the ability to reproduce the work. The paper is well written and easy to follow.”

We thank the reviewer for the encouraging comment.

“I found some of the graph (Figure 1) a bit crowded with the label of different significant level.”

We thank the reviewer for this suggestion. We modified Figure 1c, d by using only one set of significance marks on the side because most of the points on the curves are significant. We also did similar changes for Figure 6a, b and supplementary Figure 3b, d. We modified the figure legends accordingly: “Most data points of ctrl_0h, HL_4h, and HT_4h were statistically significantly different compared to ctrl_4h (*p<0.05) denoted by asterisks at the end of curves.”

Reviewer #3 (Remarks to the Author):

“I have reviewed the article entitled “High Light and High Temperature Reduce Photosynthesis via Different Mechanisms in the C₄ Model *Setaria viridis*”. Plants were grown at 250 PPF of light at 31°C in growth chamber. On 13 days after sowing, one set of plants were placed under 900 PPF and 31°C for 4 hours and that was called the high light treatment, and another set of plants were placed at 250 PPF and 40°C for 4 hours and that was called the high temperature treatment. After the treatment, gas exchange and leaf chlorophyll fluorescence were measured. Leaves also were sampled for transmission electron microscopy images, transcriptome and abscisic acid analysis.”

We thank the reviewer for providing feedback on our manuscript.

“The experiment was poorly designed and does not reproduce a real problem. First: high light is only a problem to C₄ plants at low temperature. “

We respectfully disagree as we think high light compromises C₄ photosynthesis under various stressful conditions, e.g. high temperature, drought, but not only low temperature. Although the stress combination has not been well studied in C₄ plants, it is well

documented in C₃ plants that high light exaggerates the inhibition of photosynthesis in the presence of other stresses. When photosynthesis is compromised by high temperature or drought stresses, high light further increases oxidative stress and reduces photosynthesis. See this reference below:

Balfagón, D. *et al.* Jasmonic acid is required for plant acclimation to a combination of high light and heat stress. *Plant Physiology* **181**, 1668–1682 (2019).

“At high temperature like 31°C, near the thermal optimum for most of the C₄ plants, photosynthesis is believed to be limited by the ETR.”

The regulation of photosynthesis is dynamic, depending on the environmental and developmental conditions. Consequently, it is debatable that one single factor limits photosynthesis.

“In fact, C₄ plants are noted to have a higher quantum yield than C₃ plants.”

In a classic study by Ehleringer and Björkman (1977), a comparison of C₃ and C₄ species showed that they had very similar values of quantum yield of CO₂ assimilation (Φ_{CO_2}) under the normal air condition and leaf temperature of 30°C. In contrast, above a temperature of ~30°C, this same study showed that the values of Φ_{CO_2} in a C₃ species dropped below that of the C₄ species, whereas at temperatures below 30°C, Φ_{CO_2} was slightly higher in the C₃ than C₄ species.

Ehleringer, J. & Björkman, O. Quantum yields for CO₂ Uptake in C₃ and C₄ Plants: Dependence on Temperature, CO₂, and O₂ Concentration. *Plant Physiology* **59**, 86–90 (1977).

“Second, light levels of 250 PPFD are really low to grow C₄ plants.”

Our growth light is comparable with previous publications using *S. viridis*. Here we include a list of relevant publications to support our growth light, but we also cited some of these references in the Methods section, line 594, page 20 of the previous version.

For example:

A paper from the group of Dr. Julian Hibberd: *S. viridis* growth light 200 $\mu\text{mol photons m}^{-2} \text{s}^{-1}$

<https://academic.oup.com/plphys/article/165/1/62/6113262>

John, C. R., Smith-Unna, R. D., Woodfield, H., Covshoff, S. & Hibberd, J. M. Evolutionary convergence of cell-specific gene expression in independent lineages of C₄ grasses. *Plant Physiology* **165**, 62–75 (2014).

A paper from the group of Dr. Susanne von Caemmerer: *S. viridis* growth light 200 $\mu\text{mol photons m}^{-2} \text{s}^{-1}$

<https://www.nature.com/articles/s42003-019-0561-9>

Ermakova, M., Lopez-Calcagno, P. E., Raines, C. A., Furbank, R. T. & von Caemmerer, S. Overexpression of the Rieske FeS protein of the Cytochrome b_6f complex increases C_4 photosynthesis in *Setaria viridis*. *Commun Biol* **2**, 314 (2019).

A paper from the group of Dr. Xin-Guang Zhu: *S. viridis* optimum growth light 150- 200 $\mu\text{mol photons m}^{-2} \text{s}^{-1}$

<https://www.sciencedirect.com/science/article/abs/pii/S0176161720301346>

Essemine, J. *et al.* Photosynthetic and transcriptomic responses of two C_4 grass species with different NaCl tolerance. *Journal of Plant Physiology* **253**, 153244 (2020).

A paper from the group of Dr. Ivan Baxter: *S. viridis* growth light 230 $\mu\text{mol photons m}^{-2} \text{s}^{-1}$

<https://pubmed.ncbi.nlm.nih.gov/30093527/>

Feldman, M. J. *et al.* Components of water use efficiency have unique genetic signatures in the model C_4 grass *Setaria*. *Plant Physiol* **178**, 699–715 (2018).

We cited all of these papers mentioned above in the revised version. Our *S. viridis* plants grown under the leaf level light of 250 $\mu\text{mol photons m}^{-2} \text{s}^{-1}$ were very healthy and reached full size for photosynthetic measurements in just 13 days after sowing. Thus, we believe the growth light we used in this research was valid, physiologically relevant, and our results are comparable with other experiments using *S. viridis* in the literature.

“The high light treatment of 900 PPFD is still pretty low light. In regions/seasons where temperature reaches 31°C, light level at field condition frequently overcomes 1300 PPFD. And this is actually what the data shows: that photosynthesis is limited by light! Whenever light availability was greater, photosynthesis boosted up. Thus, light levels of 250-900 PPFD are very unnatural and does not reflect what we find in nature. “

We agree with the reviewer that light at the top of a canopy could reach 2000 to 2500 $\mu\text{mol photons m}^{-2} \text{s}^{-1}$ in field conditions, but within most canopies there are very steep light gradients due to interception and absorption of radiation. Thus, many, if not most, of the leaves in canopies experience more dynamic and fluctuating light intensities than those at the top of the canopy.

Our experiments were conducted within a growth chamber with controlled conditions. We clarify that the aim of our research was not to investigate all range of light intensities or extreme high light; instead we focused on moderately high light, which was about 3~4 fold higher than the control growth light.

In field condition, *S. viridis* frequently experiences the light range of 250-900 $\mu\text{mol photons m}^{-2} \text{s}^{-1}$, especially during the early morning or late afternoon time.

We agree with the reviewer that photosynthesis increased with light in *S. viridis* leaves, however, we highlighted that HL-treated plants had lower photosynthetic capacity than the control plants (Figure 1, Figure 8, and Supplementary Figure 4).

“I do believe that there are some gene expression related to high light stress. But I believe that a lot of genes that would naturally express in field conditions did not express in growth

chamber conditions and might have been expressed after the plant finally reached light levels near field condition. Which genes are naturally expressed in the field and which genes will just express under high light stress?”

Research using either growth chamber conditions or field conditions has significant values and both approaches contribute to our understanding of plant biology. While studies under field conditions are influenced by the confounding effects of multiple environmental drivers, growth chamber experiments can control conditions and test for specific processes. The known challenge is how to reconcile and unify both approaches for a complete understanding of plant biology under changing environmental conditions.

We emphasize that our experiment was performed under controlled growth chamber conditions and our results are comparable with previous research efforts using a similar approach. Under our experimental conditions in environment-controlled growth chambers, we detected large transcriptional changes in response to HL treatment in *S. viridis* leaves. We believe that our research provides an important baseline to inform further hypotheses for C₄ plant performance under field conditions.

“Figures were also poorly designed, frequently too busy, and hard to understand. Legends are lengthy with a lot of unnecessary text.”

After carefully considering the reviewer’s comments and getting feedback from the editor, we simplified most of the figure legends in the revised version. Meanwhile, we followed the guidelines of Communications Biology to include the necessary information in the figure legends to help readers understand each figure without referring to other figures or the main text.

<https://www.nature.com/documents/commsj-life-style-formatting-guide-accept.pdf>

“Figure 7 for example should be two different figures.”

We appreciate the reviewer’s comment. However, Communications Biology requires no more than 10 main display items (including both figures and tables), so the editor recommended to retain Figure 7 as it is. Additionally, we think Figure 7 as a whole shows a complete story about how HL and HT affected chloroplast ultrastructures in *S. viridis* leaves.

Reviewers' Comments:

Reviewer #1:

Remarks to the Author:

i am fine about the revisions the authors have made.

Reviewer #2:

Remarks to the Author:

Th authors successfully addressed all questions has been raised by the reviewers. The manuscript has been improved in this revised form.